# Efficient and Scalable Density Functional Theory Hamiltonian Prediction through Adaptive Sparsity

**Erpai Luo** [* 1]  **Xinran Wei** [* 2]  **Lin Huang** [2]  **Yunyang Li** [3]  **Han Yang** [2]  **Zaishuo Xia** [4]  **Zun Wang** [2]  **Chang Liu** [2]
**Bin Shao** [2]  **Jia Zhang** [2]

## Abstract

Hamiltonian matrix prediction is pivotal in computational chemistry, serving as the foundation for determining a wide range of molecular properties. While SE(3) equivariant graph neural networks have achieved remarkable success in this domain, their substantial computational cost—driven by high-order tensor product (TP) operations—restricts their scalability to large molecular systems with extensive basis sets. To address this challenge, we introduce SPHNet, an efficient and scalable equivariant network, that incorporates adaptive SParsity into Hamiltonian prediction. SPHNet employs two innovative sparse gates to selectively constrain non-critical interaction combinations, significantly reducing tensor product computations while maintaining accuracy. To optimize the sparse representation, we develop a Three-phase Sparsity Scheduler, ensuring stable convergence and achieving high performance at sparsity rates of up to 70%. Extensive evaluations on QH9 and PubchemQH datasets demonstrate that SPHNet achieves state-of-the-art accuracy while providing up to a 7x speedup over existing models. Beyond Hamiltonian prediction, the proposed sparsification techniques also hold significant potential for improving the efficiency and scalability of other SE(3) equivariant networks, further broadening their applicability and impact. Our code can be found at https://github.com/microsoft/SPHNet.

## 1. Introduction

The Kohn-Sham Hamiltonian matrix is a fundamental quantity in computational chemistry, as it enables the prediction of all molecular properties obtainable through traditional Density Functional Theory (DFT) calculations (Hohenberg & Kohn, 1964; te Vrugt et al., 2020). Accurately predicting the Hamiltonian matrix allows for the rapid determination of system energy, HOMO-LUMO gaps, electron density, and other important properties (Fang et al., 2022; Zang et al., 2023; Batzner et al., 2022; Batatia et al., 2022; Wang et al., 2022; Li et al., 2024; Chen et al., 2023). Consequently, recent efforts have focused on using machine learning techniques to predict the Hamiltonian matrix (Schütt et al., 2017; Unke et al., 2021; Yu et al., 2023; Gong et al., 2023; Zhou et al., 2022; Zhong et al., 2023). Among these, methods based on SE(3) equivariant graph neural networks (Unke et al., 2021; Yu et al., 2023; Gong et al., 2023; Yu et al., 2023; Li et al., 2025b) have achieved superior accuracy due to their consistency with the equivariance of the Hamiltonian matrix. These methods typically project molecular features into spherical harmonics space and use tensor product operations to interact features of different orders, which are crucial for maintaining equivariance.

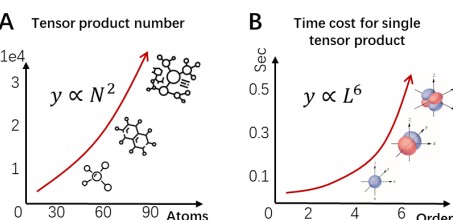

*Figure 1.* **(A)** The number of tensor products grows quadratically with the number of atoms $N$, as the Hamiltonian includes features for all possible atomic pair combinations. **(B)** The time cost of tensor products grows with the sixth power of their order $L$, where the increase in order corresponds to the expansion in the number of orbital types in the DFT basis set. For example, the def2-SVP basis set requires a maximum order of 4, while def2-TZVP demands an order of 6.

However, the efficiency issues introduced by tensor product operations have become a significant bottleneck. On the one hand, as the number of atoms in the system increases, the

---

[*]Equal contribution  [1]Department of Automation, Tsinghua University, Beijing, China [2]Microsoft Research AI for Science, Beijing, China [3]Department of Computer Science, Yale University, CT, USA [4]Department of Computer Science, University of California, CA, USA. Correspondence to: Xinran Wei <weixinran@microsoft.com>, Lin Huang <huang.lin@microsoft.com>, Jia Zhang <jia.zhang@microsoft.com>.

*Proceedings of the $42^{nd}$ International Conference on Machine Learning*, Vancouver, Canada. PMLR 267, 2025. Copyright 2025 by the author(s).

number of tensor product operations grows quadratically, significantly increasing computational cost and memory requirements, as illustrated in Fig.1(A). On the other hand, since each order in the output representation corresponds to different orbitals in the DFT basis set, the order of tensor product operations must increase as the DFT basis set expands. As shown in Fig. 1(B), the computational complexity of tensor products scales with the sixth power of the order, causing the time cost of a single tensor product operation to grow rapidly with larger basis sets. Therefore, optimizing both the number and efficiency of tensor product operations is critical for extending Hamiltonian prediction to larger systems and larger basis sets.

The recent advancements in leveraging sparsity in numerical DFT solvers (Burow & Sierka, 2011; Laqua et al., 2022) and message-passing networks (Liu et al., 2023; Peng & Zhang, 2022; Passaro & Zitnick, 2023) inspire our exploration of incorporating adaptive sparsity into Hamiltonian prediction models. To address the aforementioned challenges, we developed SPHNet, and the main contribution of this work can be summarized as follows:

- We propose a scalable and efficient network for Hamiltonian prediction, termed **SPHNet**, which introduce adaptive sparsity into equivariant networks by introducing two sparse gates. Specifically, we employ the **Sparse Pair Gate** to filter out unimportant node pairs, reducing the number of tensor product computations, and the **Sparse TP Gate** to prune less significant interactions across different orders in tensor product, thereby improving the efficiency of tensor operations.

- To optimize the sparse representation, we develop a **Three-phase Sparsity Scheduler**, which ensures efficient weight updates for all combinations through three stages: random, adaptive, and fixed. This approach facilitates stable convergence to the optimal set while maintaining high accuracy at sparsity rates up to 70%.

- We demonstrate the enhancements of SPHNet across multiple datasets, showing that it outperforms existing models in accuracy on QH9 and PubchemQH, while achieving up to a 7x speedup and reducing memory usage by up to 75%. It also demonstrates comparable performance on small molecular trajectories on the MD17 dataset. Moreover, the proposed adaptive sparsification techniques exhibit strong promise in improving the computational efficiency and scalability of other SE(3) equivariant networks, paving the way for broader applications across diverse tasks.

## 2. Related Works

**SE(3) Equivariant Neural Network.** The SE(3) equivariant neural network is widely used in AI for chemistry

due to its unique advantage in predicting quantum tensors (Fuchs et al., 2020; Du et al., 2022; Musaelian et al., 2023; Liao & Smidt, 2022; Liao et al., 2023; Batzner et al., 2022; 2023). Key models include SE(3)-Transformer (Fuchs et al., 2020), which introduced a robust self-attention mechanism for 3D point clouds and graphs, and Equiformer (Liao & Smidt, 2022), which predicted molecular properties using SE(3) Transformer architecture. EquiformerV2 (Liao et al., 2023) improved on this by employing efficient eSCN convolutions (Passaro & Zitnick, 2023), outperforming traditional networks like GemNet (Gasteiger et al., 2022) and Torchmd-Net (Thölke & De Fabritiis, 2022). Allegro (Musaelian et al., 2023) used a local, equivariant model without atom-centered message passing, showing excellent generalization.

**Hamiltonian Matrix Prediction.** Hamiltonian prediction has advanced with neural networks in recent years. SchNOrb (Schütt et al., 2019) extended SchNet (Schütt et al., 2017) for high-accuracy molecular orbital predictions, while PhiSNet (Unke et al., 2021) successfully introduced SE(3) networks, significantly improving accuracy while facing inefficiency due to tensor product operations. QHNet (Yu et al., 2023) improved efficiency by reducing the number of tensor products and (Li et al., 2025b) further extends the scalability and applicability of Hamiltonian matrix prediction on large molecular systems by introducing a novel loss function. At the same time, methods such as DeepH (Li et al., 2022) are focused on the Hamiltonian prediction of the periodic system.

**Network Sparsification.** Network sparsification enhances computational efficiency by removing redundant neural network components. Early methods like Optimal Brain Damage (LeCun et al., 1990) and Optimal Brain Surgeon (Hassibi & Stork, 1993) pruned weights based on importance, followed by retraining to restore performance. Iterative pruning and retraining (Han et al., 2015) became widely used for reducing complexity while preserving accuracy. The Lottery Ticket Hypothesis (LTH) (Frankle & Carbin, 2019) later showed that sparse subnetworks in dense models could independently match full model performance. Sparsification has since expanded to target weights, nodes, and edges, proving effective in molecular property prediction (Liu et al., 2023; Peng & Zhang, 2022). Recent work has also extended to pruning weights of the Clebsch-Gordan tensor product (Wang et al., 2023), demonstrating potential despite unproven efficiency.

## 3. Preliminary

**DFT Hamiltonian**. Density Functional Theory (DFT), as formulated by Hohenberg and Kohn (Hohenberg & Kohn, 1964) and further developed through the Kohn-Sham equations by Kohn and Sham (Kohn & Sham, 1965), represents a key computational quantum mechanical methodology for

studying the electronic structure of many-body systems, including atoms, molecules, and solids (Jain et al., 2016; te Vrugt et al., 2020). The core principle of DFT lies in solving the matrix $\mathbf{C}$ via the Kohn-Sham (KS) equations, summarized briefly as (Szabo & Ostlund, 2012):

$$\mathbf{HC} = \mathbf{SC}\boldsymbol{\epsilon}, \qquad (1)$$

where $\mathbf{C}$ representing the coefficients of molecular orbitals, $\mathbf{H}$ signifies the Hamiltonian matrix, $\mathbf{S}$ refers to the overlap matrix, and $\boldsymbol{\epsilon}$ is the diagonal matrix encapsulating orbital energies. Note that $\mathbf{C} \in \mathbb{R}^{n \times n_o}$, $\mathbf{H}, \mathbf{S} \in \mathbb{R}^{n \times n}$, and $\boldsymbol{\epsilon} \in \mathbb{R}^{n_o \times n_o}$, where $n$ denotes the number of basis functions and $n_o$ denotes the number of atomic orbitals used in DFT calculations. Primarily, an iterative self-consistent field (SCF) methodology (Payne et al., 1992; Cances & Le Bris, 2000; Wang & Baerends, 2022; Kudin et al., 2002) is utilized to solve the Kohn-Sham (KS) equations and to obtain the Hamiltonian matrix, requiring a temporal complexity of $\mathcal{O}(n^3)$.

**SE(3) Equivariant and Tensor Product.** The coordinate system in 3D Euclidean space is the most commonly used description for molecular systems. In this space, the group of 3D translations and rotations forms the SE(3) group. A function $f(\cdot)$ is SE(3) equivariant if it satisfies $f(D(g)x) = D(g)f(x)$, where $D(g)$ is a representation of SE(3) acting on $x$. Therefore, a neural network in which every operation satisfies the definition of $f(\cdot)$ is referred to as an SE(3) equivariant neural network. Among these operations, the tensor product operation is the most crucial. Typically, higher-order irreducible representations (irreps) are constructed using spherical harmonics. The tensor product operation combines two irreps, $x$ and $y$, with rotational orders $\ell_1$ and $\ell_2$ respectively, using the Clebsch-Gordan (CG) coefficients (Griffiths & Schroeter, 2018), resulting in a new irreducible representation of order $\ell_3$ as:

$$(x^{\ell_1} \otimes y^{\ell_2})_{m_3}^{\ell_3} = \sum_{m_1=-\ell_1}^{\ell_1} \sum_{m_2=-\ell_2}^{\ell_2} C_{(\ell_1,m_1),(\ell_2,m_2)}^{(\ell_3,m_3)} x_{m_1}^{\ell_1} y_{m_2}^{\ell_2},$$

$$\qquad (2)$$

where $m$ denotes the $m$-th element in the irreducible representation and satisfies $-\ell \leq m \leq \ell$. The output order $\ell_3$ is restricted by $|\ell_1 - \ell_2| \leq \ell_3 \leq |\ell_1 + \ell_2|$. Since each order has to interact with every other order, the computational complexity of the full tensor product using irreps up to order L have a computational complexity of $O(L^6)$, which constrains its applicability for systems with higher degrees.

# 4. Methodology

## 4.1. Three-phase Sparsity Scheduler

To ensure stable selection of combinations in the Sparse Tensor Product Gate and Sparse Pair Gate, we propose the

Three-phase Sparsity Scheduler, a three-stage selection function that always selects retained elements from the set based on a learnable weight matrix $\mathbf{W}$. Thus, for any unsparsified set $U$, the scheduler can be defined as:

$$\mathrm{TSS}(\mathbf{W}, k) = \begin{cases} \mathrm{RANDOM}(\mathbf{W}, 1-k), & \text{if epoch} < t, \\ \mathrm{TOP}(\mathbf{W}, 1-k), & \text{if epoch} = t, \\ \mathrm{TOP}(\mathbf{W}^{p_2\prime}, 1-k), & \text{if epoch} > t, \end{cases}$$

$$\qquad (3)$$

where $k$ represents the sparsity rate of the tensor product, $t$ is the round of the training epoch, and $\mathrm{TSS}(\cdot)$ always returns a subset $U^{\mathrm{TSS}}$ of $U$, containing $(1-k)|U|$ elements selected based on $\mathbf{W}$.

Specifically, in the first phase, $\mathbf{W}$ is initialized as an all-ones vector, and $\mathrm{RANDOM}(\cdot)$ denotes a random function that selects elements from the set independently of $\mathbf{W}$ with a given probability $1-k$. This ensures that all parameters are unbiasedly selected and updated. In the second phase, $\mathrm{TOP}(\cdot)$ is used to select the elements whose weights are within the top $1-k$ percent of all elements, ensuring that the optimal combinations are chosen globally. In the third phase, $\mathbf{W}^{p_2\prime}$ represents a fixed vector without gradients, which is frozen after the final backward update of the second phase. From this point onward, the selected combinations remain unchanged. This design prioritizes efficiency, as static connections typically result in faster computation.

## 4.2. Sparse Tenor Product Gate

The Sparse TP Gate is designed to accelerate individual tensor product computations by filtering out unnecessary cross-order combinations in traditional tensor product operations, as illustrated in Fig.2(B). By doing so, tensor product operations implemented with e3nn (Geiger et al., 2022) can achieve near-linear acceleration, as demonstrated in the Appendix B.3. Moreover, this approach can be easily adapted to various tensor product implementations by simply modifying the instructions that declare cross-order combinations in the tensor product operation.

Generally, the classical tensor product operation includes the interaction of every element $m$ in every irrep order $\ell$. The aim of Sparse Tensor Product Gate is to learn a set of the most valuable combinations $U_c^{\mathrm{TSS}}$ from the complete set $U_c = \{(\ell_1, \ell_2, \ell_3) \mid \ell_3 \in [|\ell_1 - \ell_2|, \ell_1 + \ell_2]\}$ in the tensor product and remove the redundant combinations accordingly. $U_c^{\mathrm{TSS}}$ is defined by a one-dimensional learnable value score $\mathbf{W}_c^{\ell_1,\ell_2,\ell_3}$ of length $|U_c|$ as:

$$U_c^{\mathrm{TSS}} = \{(\ell_1, \ell_2, \ell_3) \mid \mathbf{W}_c^{\ell_1,\ell_2,\ell_3} \in \mathrm{TSS}(\mathbf{W}_c, k)\}, \quad (4)$$

where $\mathrm{TSS}(\cdot)$ is the selection scheduler defined in Equation 3. Accordingly, we can obtain the updated combination weight from the original weights $c$ defined in classical tensor

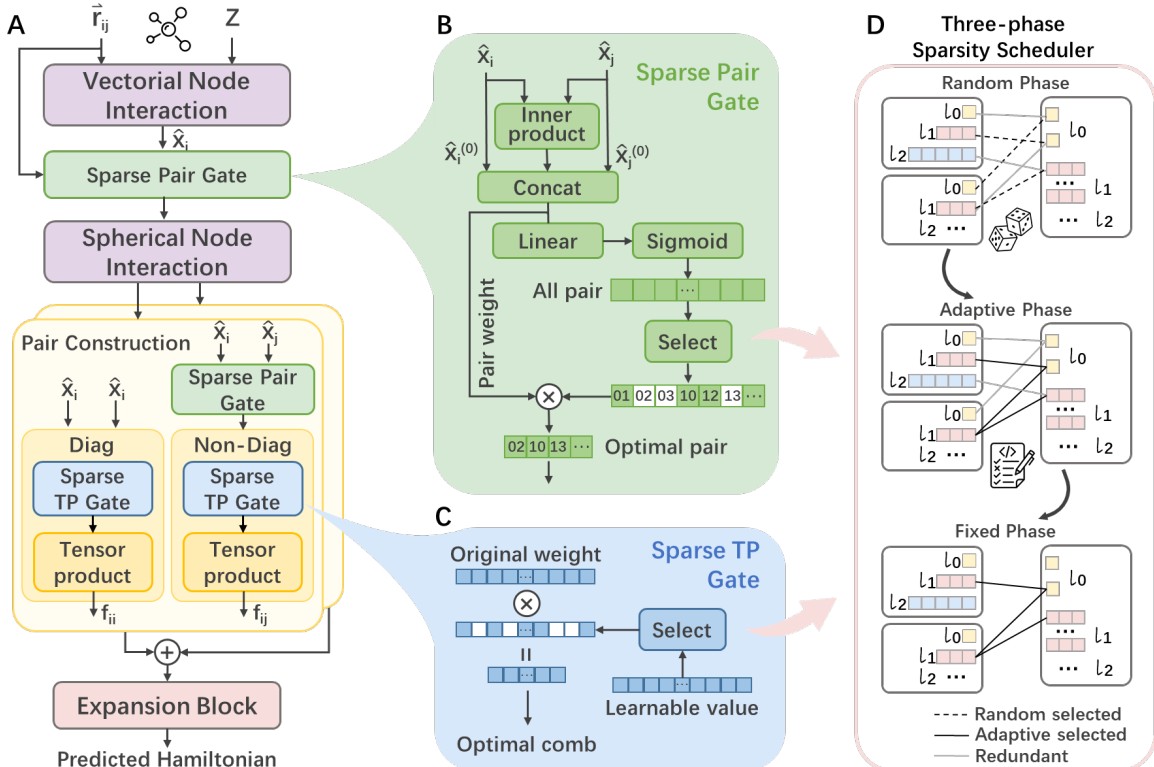

*Figure 2.* **(A)** The overall architecture of SPHNet. Atomic numbers and atomic coordinates are first passed through the Vectorial Node Interaction Blocks to obtain atomic features $\mathbf{x}_i^\ell$. Subsequently, the Sparse Pair Gate selects the key pair set $(i, j)$ for the Spherical Node Interaction Blocks, where the irreps $\mathbf{x}_i^\ell$ are elevated from the $\ell = 1$ to $L_{\max}$ during the interaction process. Next, the Sparse Tensor Product Gate in the construction block identifies the key cross-order combinations $(\ell_1, \ell_2, \ell_3)$ for the diagonal blocks, yielding diagonal pair features $\mathbf{f}_{ii}$. For non-diagonal blocks, both the Pair Gate and Tensor Product Gate are applied to select the critical pairs and tensor product combinations, producing non-diagonal pair features $\mathbf{f}_{ij}$. Finally, these features are fed into the expansion block to construct the predicted Hamiltonian matrix. **(B)** Sparse Pair Gate: It takes pairwise features as input, computes weights for each pair $(i, j)$, and selects a optimal subset using the sparsity scheduler. **(C)** Sparse Tensor Product Gate: Similarly, it utilizes the sparsity scheduler to identify an optimal subset of cross-order combinations $(\ell_1, \ell_2, \ell_3)$ based on learnable weights. **(D)** Three-Phase Sparsity Scheduler: Designed for the sparse gates, it operates in three phases: random, adaptive, and fixed.

product by:

$$c' = \{c^{\ell_1, \ell_2, \ell_3} \times \mathbf{W}_c^{\ell_1, \ell_2, \ell_3} \mid (\ell_1, \ell_2, \ell_3) \in U_c^{\mathrm{TSS}}\}. \quad (5)$$

In practical experiments, the sparsity rate $k$ is set based on the details in Section 5.4.

### 4.3. Sparse Pair Gate

Since the output of the Hamiltonian matrix encompasses interactions between all pairs of atoms, previous models typically define inter-atomic relationships as fully connected. This results in tensor product computations that scale quadratically with the number of nodes. However, we observed that not every block requires such dense connectivity. To reduce the number of tensor product operations in the network, we developed the Sparse Pair Gate, which adaptively selects the most important edges.

Similarly with the tensor product gate, the Sparse Pair Gate

selects a subset from all the node pair $U_p = \{(i, j) | i \neq j\}$, and uses a linear layer $F_p(\cdot)$ to learn the weight of each pair $\mathbf{W}_p^{ij}$ in this fully connected graph. This can be formulated as below:

$$\mathbf{I}_{ij} = (\mathbf{x}_i^0 \| \mathbf{x}_j^0 \| \langle \mathbf{x}_i, \mathbf{x}_j \rangle^{1:}), \quad (6)$$

$$\mathbf{W}_p^{ij} = \mathrm{Sigmoid}\left(F_p(\mathbf{I}_{ij})\right), \quad (7)$$

where $\mathbf{x}_i^0$ and $\mathbf{x}_j^0$ are the zero-order features of the irreps, $\|$ is concatenation operation in the last feature dimension and $\langle \cdot, \cdot \rangle$ stands for the inner product operation. Thus, the sparse pair gate can be formulated as:

$$U_p^{\mathrm{TSS}} = \{(i, j) \mid \mathbf{W}_p^{ij} \in \mathrm{TSS}(\mathbf{W}_p, k)\}, \quad (8)$$

where $\mathrm{TSS}(\cdot)$ is the same as Equation 3. Accordingly, we can get the pair weight $\mathbf{w}_{ij}$ of $\mathbf{x}_i, \mathbf{x}_j$ used in tensor product:

$$\mathbf{w}_{ij} = F_r\left(\mathrm{RBF}(\vec{r}_{ij})\right) \times F_s(\mathbf{W}_p^{ij} \times \mathbf{I}_{ij}), \quad (9)$$

where $F_r(\cdot)$ and $F_s(\cdot)$ are linear layers, $\mathrm{RBF}(\cdot)$ is the radial basis function. The extra computational cost introduced by above sparse gates and the scheduler is minimal and has negligible impact on the overall time, which is discussed in detailed in Appendix D.7.

## 4.4. SPHNet

With the above two sparse gates, SPHNet can be formulated as four modules. The model takes atomic numbers $Z$ and 3D coordinates of molecular systems as input features, initializes the node representation $x_i$ with a linear function as $x_i = F_x(Z_i)$, processes them through the Vectorial Node Interaction Block, and sequentially passes through the Spherical Node Interaction Block, Pair Construction Module, and Expansion Module to output the predicted Hamiltonian matrix $\mathbf{H}$, as illustrated in Fig.2.

**Node Interaction Block**. The Node Interaction Block aggregates information from neighboring nodes through a message-passing mechanism, thereby extracting irreducible representations of nodes. Specifically, the irreducible representation $x_i$ can be updated by aggregating the message $m_{ij}$ as,

$$\hat{\mathbf{x}}_i^\ell = F_m(\mathbf{x}_i^\ell + \sum_j \mathbf{m}_{ij}^\ell), \quad (10)$$

where $F_m(\cdot)$ is a linear layer.

To achieve sufficiently interactive high-order information while minimizing the associated computational cost, we propose a mechanism that gradually increases the maximum order of node representations. Initially, four Vectorial Node Interaction Blocks are applied to obtain vectorial representations $\mathbf{m}_{ij}^\ell$, where $\ell \leq 1$, as follows:

$$\mathbf{m}_{ij}^\ell = \mathbf{x}_j^\ell \odot (\mathbf{w}_{ij} \odot \vec{r_{ij}}), \quad \text{s.t.} \quad l \leq 1, \quad (11)$$

where $\mathbf{w}_{ij}$ is the weight defined in Equation 9, and $\vec{r_{ij}}$ is the distance vector between atoms. This representation does not involve interactions between high-order tensors, thereby eliminating tensor product operations, as detailed in Appendix D.3.

Next, we apply two Spherical Node Interaction Blocks to capture interactions for high-order irreducible representations, increasing the highest order of the irreducible representation to match the maximum required order determined by the basis set. Specifically, these blocks project distance information $\vec{r_{ij}}$ into the high-order spherical space using spherical harmonics function $Y_m^\ell(\cdot)$. Subsequently, $\mathbf{m}_{ij}^\ell$ in this block is computed under the constraints imposed by Sparse Pair Gate as:

$$\mathbf{m}_{ij}^{\ell_3} = \sum_{\ell_1,\ell_2} \mathbf{x}_j^{\ell_2} \otimes_{\ell_1,\ell_2}^{\ell_3} \mathbf{w}_{ij}^{\ell_1,\ell_2,\ell_3} Y_m^{\ell_2}(\vec{r_{ij}}),$$
$$\text{s.t.} \quad (i,j) \in U_p^{\mathrm{TSS}}, \quad \ell_1, \ell_2, \ell_3 \leq L_{\max}, \quad (12)$$

where $L_{\max}$ represents the allowed max order of irreps in the network and $\mathbf{w}_{ij}$ is the weight defined in Equation 9.

**Pair Construction Block**. Consists of the Non-Diagonal block and the Diagonal block. The Non-Diagonal block uses these irreducible representations to compute the interaction from node $j$ to node $i$, generating the pair-wise features $\mathbf{f}_{ij}$ for the Non-Diagonal blocks of the Hamiltonian matrix. The Diagonal block calculates the self-interaction of node $i$ to produce the node-wise feature $\mathbf{f}_{ii}$ for the diagonal blocks of the Hamiltonian matrix.

As shown in Fig.2.(B), we use Sparse Tensor Product Gate before both the Diagonal and Non-Diagonal block, and use Sparse Pair Gate before Non-Diagonal block, therefore the $\mathbf{f}_{ii}$, $\mathbf{f}_{ij}$ can be reformulated as:

$$\mathbf{f}_{ii}^{\ell_3} = \sum_{\{\ell_1,\ell_2,\ell_3\} \in U_c^{\mathrm{TSS}}} \mathbf{W}_{ii}^{(\ell_1,\ell_2,\ell_3)} \times (\hat{\mathbf{x}}_i^{\ell_1} \otimes_{\ell_1,\ell_2}^{\ell_3} \hat{\mathbf{x}}_i^{\ell_2}), \quad (13)$$

$$\mathbf{f}_{ij}^{\ell_3} = \sum_{\{\ell_1,\ell_2,\ell_3\} \in U_c^{\mathrm{TSS}}} c'^{(\ell_1,\ell_2,\ell_3)} \times (\hat{\mathbf{x}}_i^{\ell_1} \otimes_{\ell_1,\ell_2}^{\ell_3} \mathbf{w}_{ij}^{\ell_1,\ell_2,\ell_3} \hat{\mathbf{x}}_j^{\ell_2}),$$
$$\text{s.t.} \quad (i,j) \in U_p^{\mathrm{TSS}}, \quad \ell_1, \ell_2, \ell_3 \leq L_{\max}, \quad (14)$$

where $W_{ii}$ is learnable parameters for each combination in $U_c^{\mathrm{TSS}}$, $c'$ is an updated combination weight defined in Equation 5 and $\mathbf{w}_{ij}$ is the weight defined in Equation 9. It's noteworthy that the Sparse Pair Gate is only employed from the second Spherical Node Interaction Block and the Non-Diagonal Pair Construction block to ensure complete information flow between nodes, and more details could be found in Appendix D.5.

**Expansion Block**. Finally, the expansion block utilizes the tensor expansion operation to obtain the non-diagonal Hamiltonian block $\mathbf{h}_{ij}$ and diagonal Hamiltonian block $\mathbf{h}_{ii}$. The detailed formulation can be found in Appendix D.6. Considering the symmetry of the Hamiltonian matrix, where for any sub-block $\mathbf{h}_{ij}$, there exists a corresponding sub-block $\mathbf{h}_{ji}$ such that $\mathbf{h}_{ji} = \mathbf{h}_{ij}^T$. So we can further enhance the model's efficiency by constructing the node pair features $\mathbf{f}_{ij}$ from the upper triangular part of the Hamiltonian matrix, i.e., $\{\mathbf{f}_{ij} \mid i < j\}$. Leveraging this symmetry allows us to halve the number of tensor products required when predicting the node pair irreps $\mathbf{f}_{ij}$, significantly reducing the computational burden associated with constructing $\mathbf{h}_{ij}$.

## 5. Result

We conducted a series of experiments to compare the overall performance of SPHNet with the previous state-of-the-art model, including SchNOrb (Schütt et al., 2019), PhiSNet (Unke et al., 2021), QHNet (Yu et al., 2023), and WANet (Li et al., 2025b), and the results listed in the table are sourced from their papers. Note that the results for PhiSNet and SchNOrb can only be conducted on the MD17 dataset, as these models are specifically designed for trajectory datasets

Table 1. Experimental results on QH9 dataset.

| Dataset | Model | $H \downarrow$ $[10^{-6}E_h]$ | $\epsilon \downarrow$ $[10^{-6}E_h]$ | $\psi \uparrow$ $[10^{-2}]$ | Memory $\downarrow$ [GB/Sample] | Speed $\uparrow$ [Sample/Sec] | Speedup $\uparrow$ Ratio |
|---|---|---|---|---|---|---|---|
| QH9-stable iid | QHNet | 76.31 | 798.51 | 95.85 | 0.70 | 19.2 | 1.0x |
| | WANet | 80.00 | 833.62 | 96.86 | N/A | N/A | N/A |
| | SPHNet | **45.48** | **334.28** | **97.75** | **0.23** | **76.80** | **4.0x** |
| QH9-stable ood | QHNet | 72.11 | 644.17 | 93.68 | 0.70 | 21.12 | 1.0x |
| | SPHNet | **43.33** | **186.40** | **98.16** | **0.23** | **78.72** | **3.7x** |
| QH9-dynamic geo | QHNet | 70.03 | 408.31 | 97.13 | 0.70 | 24.68 | 1.0x |
| | WANet | 74.74 | 416.57 | **99.68** | N/A | N/A | N/A |
| | SPHNet | **52.18** | **100.88** | 99.12 | **0.23** | **82.56** | **3.3x** |
| QH9-dynamic mol | QHNet | 120.10 | 2182.06 | 89.63 | 0.70 | 23.04 | 1.0x |
| | SPHNet | **108.19** | **1724.10** | **91.49** | **0.23** | **80.64** | **3.5x** |

in conformational space. Additionally, since WANet has not been open-sourced, we only report results on a subset of datasets and metrics based on the information provided in their paper. Detailed experimental settings are described in Appendix A.

**Datasets.** Three datasets were used in the experiments: The MD17 dataset includes Hamiltonian matrices for the trajectory data of four molecules: water, ethanol, malondialdehyde, and uracil, containing 4,900, 30,000, 26,978, and 30,000 structures, respectively. The QH9 dataset (Yu et al., 2024), consisting of Hamiltonian matrices for 134k small molecules with no more than 20 atoms, and the Pub-ChemQH dataset (Li et al., 2025b), containing Hamiltonian matrices for 50k molecules with atom counts ranging from 40 to 100. The QH9 and PubChemQH use the B3LYP exchange-correlation function, and the MD17 uses the PBE exchange-correlation function. The orbital basis Def2-SVP is used for MD17 and QH9, while the Def2-TZVP is used for PubChemQH. Compared to the MD17 and QH9 datasets, the maximum number of orbitals in the Hamiltonian of the PubChemQH dataset increases by about 15 times, resulting in a 200x difference in the number of elements in the Hamiltonian matrix.

**Evaluation metrics.** The evaluation metrics include Hamiltonian MAE (mean absolute error between predicted and Hamiltonian matrices calculated by DFT), occupied energy MAE $\epsilon$ (mean absolute error of energies of occupied molecular orbitals calculated from predicted Hamiltonian matrices), C similarity(cosine similarity of coefficients for occupied molecular orbitals), training GPU memory (GB per sample), training speed (training samples per second) and speedup ratio compared to QHNet.

### 5.1. Performance on QH9 Dataset

We first evaluated the performance of SPHNet on the QH9 dataset. Note that the QH9 dataset has two subsets and four different splits, including the stable-iid, stable-ood, dynamic-geo, and dynamic-mol. We trained SPHNet on

four different split sets and compared the results with the baseline models QHNet and WANet. As shown in Table 1, for two stable split sets, SPHNet attained a speedup of 3.7x to 4x over QHNet while improving the accuracy of the Hamiltonian Mean Absolute Error (MAE) and the occupied energy MAE. Furthermore, SPHNet significantly reduced GPU memory usage, requiring only 30% memory usage of the baseline model. For two dynamic split sets, SPHNet maintained a speedup of 3.3x to 3.5x, simultaneously enhancing prediction accuracy and decreasing GPU memory usage by over 70%. These results underscore SPHNet's efficiency, achieving substantial speedups and resource savings without sacrificing precision, indicating that SPHNet can effectively learn molecular features and the Hamiltonian matrix within the small-scale dataset, establishing a solid foundation for calculations in larger-scale systems.

### 5.2. Performance on PubChemQH Dataset

We further validated our approach on the larger-scale molecular system and trained SPHNet on the PubChemQH dataset. It was observed that SPHNet trains over 7.1x times faster than the baseline QHNet while achieving a better Hamiltonian prediction accuracy, as shown in Table 2, which has a higher speedup ratio compared to that in the smaller-scale datasets. Moreover, GPU memory consumption of SPHNet was significantly lower, at only 25% and 37% of the baselines.

Table 2. Experimental results on PubChemQH dataset.

| Model | QHNet | WANet | SPHNet |
|---|---|---|---|
| $H [10^{-6}E_h] \downarrow$ | 123.74 | 99.98 | **97.31** |
| $\epsilon [E_h] \downarrow$ | 3.33 | **1.17** | 2.16 |
| $\psi [10^{-2}] \uparrow$ | 2.32 | **3.13** | 2.97 |
| Memory [GB/Sample] $\downarrow$ | 22.5 | 15.0 | **5.62** |
| Speed [Sample/Sec] $\uparrow$ | 0.44 | 1.09 | **3.12** |
| Speedup Ratio $\uparrow$ | 1.0x | 2.4x | **7.1x** |

We observed that SPHNet achieves a higher speedup on

the PubChemQH dataset. As described in the dataset section, the number of orbitals in the Hamiltonians of PubChemQH is 15 times greater than in QH9. Additionally, since PubChemQH was computed using the Def2-TZVP orbital basis, the network requires a maximum output angular momentum order of $L_{\max} = 6$, whereas datasets computed with the Def2-SVP basis only require $L_{\max} = 4$. These make the number and time consumption of tensor products much higher than that in small-scale datasets. Thus, we can reasonably infer that the higher speedup achieved on the PubChemQH dataset is due to the markedly greater computational complexity of its molecules compared to those in the MD17 and QH9 datasets. This increased complexity introduces higher information redundancy, making it more amenable to adaptive sparsification. Our sparsity ablation experiments across different datasets support this inference. As molecular size and maximum angular momentum order increase, the rising sparsity rates of sparse pair gates and sparse tensor product gates generally result in smaller accuracy losses. Therefore, higher sparsity rates can be applied to SPHNet, further accelerating its performance on large-scale datasets. For more details on this ablation study, please refer to Section 5.4.

*Table 3.* Experimental results on MD17 dataset.

| Dataset | Model | $H\,[10^{-6}E_h]\downarrow$ | $\epsilon\,[10^{-6}E_h]\downarrow$ | $\psi\,[10^{-2}]\uparrow$ |
|---|---|---|---|---|
| | SchNOrb | 165.4 | 279.3 | **100.00** |
| | PhiSNet | 15.67 | 85.53 | **100.00** |
| Water | QHNet | **10.79** | **33.76** | 99.99 |
| | SPHNet | 23.18 | 182.29 | **100.00** |
| | SchNOrb | 187.4 | 334.4 | **100.00** |
| | PhiSNet | **20.09** | 102.04 | 99.81 |
| Ethanol | QHNet | 20.91 | **81.03** | 99.99 |
| | SPHNet | 21.02 | 82.30 | **100.00** |
| | SchNOrb | 191.1 | 400.6 | 99.00 |
| Malon- | PhiSNet | 21.31 | 100.6 | 99.89 |
| aldehyde | QHNet | 21.52 | **82.12** | 99.92 |
| | SPHNet | **20.67** | 95.77 | **99.99** |
| | SchNOrb | 227.8 | 1760 | 90.00 |
| | PhiSNet | **18.65** | 143.36 | 99.86 |
| Uracil | QHNet | 20.12 | **113.44** | 99.89 |
| | SPHNet | 19.36 | 118.21 | **99.99** |

## 5.3. Performance on MD17 Dataset

We also evaluated the performance of SPHNet on the smaller-scale MD17 dataset. As presented in Table 3, we compared SPHNet's performance with four baseline models across four MD17 molecules—water, ethanol, malondialdehyde, and uracil—comprising 3 to 12 atoms. The results demonstrate that SPHNet achieves accuracy comparable to other models, indicating its suitability for small molecular trajectory datasets.

It is worth noting that predictions on the MD17 dataset represent a relatively simple task compared to other datasets, as it focuses solely on small systems and their conformational

spaces. On the one hand, baseline models generally already perform well across all these datasets, leaving limited room for SPHNet to achieve significant improvements in either accuracy or speed. On the other hand, taking the water molecule with only three atoms as an example, the number of possible interaction combinations within the system is inherently small, which limits the potential benefits of adaptive sparsification.

## 5.4. Ablation Study on Sparsity Rate

The key idea of SPHNet is the implementation of adaptive sparsity to reduce computational demands in SE(3) equivariant networks. The most critical hyperparameter in this adaptive sparse system is the sparsity rate. To determine the optimal sparsity rate and assess its impact across various molecular systems, we conducted a series of experiments. We varied the sparsity rate from 0% to 90% in 10% intervals and trained SPHNet on three datasets representing different molecular scales (Here we chose Ethanol to represent MD17 dataset.). As illustrated in Fig.3, the predicted Hamiltonian MAE across all three datasets remained relatively stable at lower sparsity rates but increased significantly when the rate reached a specific threshold. This finding suggests that an appropriate range of sparsity has minimal impact on model accuracy while simultaneously enhancing computational speed.

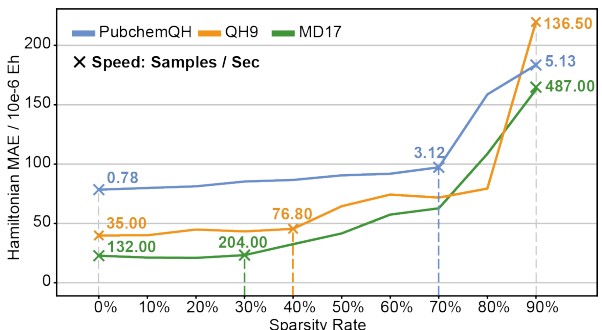

*Figure 3.* The effect of different sparsity rates on the model performance on the datasets with different scale of molecules. See the detailed computational cost scaling in Appendix B.2 of PubChemQH Dataset.

Additionally, we observed that the threshold at which the MAE begins to rise significantly becomes larger as the complexity of the molecular system increases. For the smallest MD17 system, the critical turning point is at 30%, while for the QH9 system, it is 40%, and for the largest PubChemQH system, it rises to 70%. In the experiments conducted in this study, we adopted the above critical turning point as the sparsity rate for training and testing for different datasets. This observation aligns with our earlier conclusion that the adaptive sparsity strategy offers greater potential in large-scale systems, as there are more calculations of lower importance that can be optimized. Consequently, the system size can

serve as a rough estimate for determining the applicable sparsity rate, thereby reducing the need for extensive parameter searches.

## 5.5. Analysis of Optimal Selected Set

Experiments demonstrated that the two adaptive sparse gates effectively select the most important tensor products and their optimal combinations. Here, we examined the selected pairs and combinations to understand the learning outcomes of the sparse gates.

For the optimal pair set, we analyzed the length(atomic distance) of the selected pairs, as illustrated in Fig.4(A). The results indicate that sparse gating differs from the commonly used RBF-based cutoff strategy, as it evaluates the importance of a pair based solely on its contribution to the output irreps, rather than its distance. Interestingly, as the pair length increases, the probability of a pair being selected also rises. This is likely because the features of long-range pairs are more challenging to learn, requiring as many samples as possible to achieve accurate representations. Notably, the selection probability increases nearly linearly in the distance range of 16Å to 25Å. This range corresponds to the influence of electrostatic interactions and may also include weak van der Waals forces. These findings highlight the importance of long-range interactions in Hamiltonian prediction and demonstrate the effectiveness of sparse pair gating in dynamically integrating both long-range and short-range interactions, ultimately enabling the model to achieve superior performance.

Then, for the selected optimal combination set, we analyze the order of output irreps of each selected cross-order combination in tensor product operations. For a classical tensor product operation, each irreps of the output $x^{\ell_3}$ is contributed by $O(\ell_3{}^2)$ cross-channel combinations. As shown in Fig.4(B) and Fig.7, in the QH9 dataset, the proportion of each order that is selected decreases monotonically as the order increases in both Sparse Combination Gates. In the PubChemQH, the trend is the same except the order zero has a relatively lower selected ratio. These results suggest that although higher orders' output irreps contain more combinations, the importance of a single combination is weakened as the number of combinations increases, making them easier to filter out.

## 5.6. Scaling with Increasing Size of Hamiltonian

To further validate the limits of Hamiltonian prediction scale achievable by different models under restricted GPU memory, as well as the scaling of speed and memory consumption with increasing system size, we randomly selected 6 molecules of varying sizes from the PubChemQC PM6 dataset. We computed the labels for these molecules with the setting of PubChemQH, and conducted tests with a single A6000 card based on the outcomes. The results are illustrated in Fig.5 and Fig.8.

Our method demonstrates better scaling than the baseline model. As the Hamiltonian size (number of atomic orbitals) increases, the speedup ratio improves while memory consumption decreases. Thanks to significant savings in both memory and time, our model can train on larger systems (around 3000 atomic orbitals) within the same memory consumption, whereas the baseline model is limited to approximately 1800 atomic orbitals.

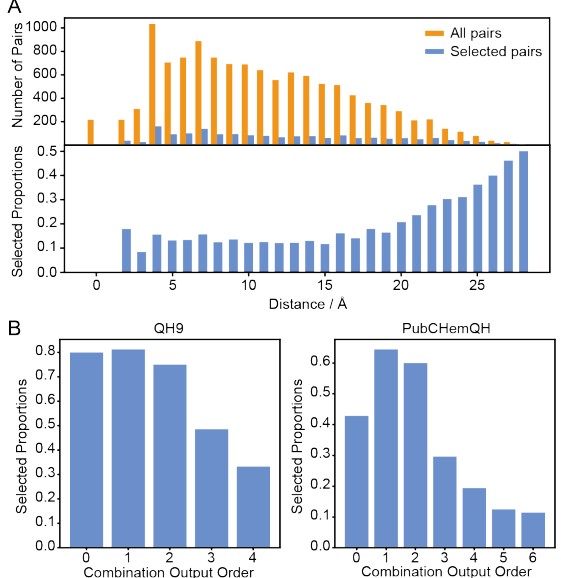

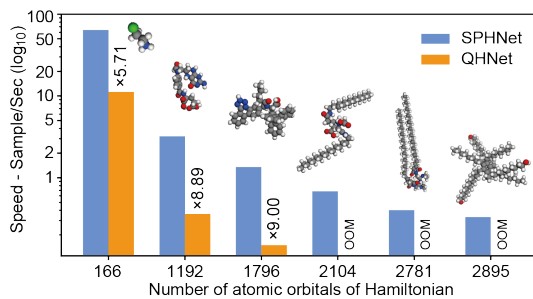

*Figure 5.* Comparison of the training speed with the increasing size of Hamiltonian.

## 5.7. Additional Ablation study on SPHNet

To evaluate the effectiveness of the Three-phase Sparsity scheduler and the Sparse Gate on the model performance, we carried out a series of ablation studies. The results showed that these strategies have significant improvements in model efficiency and accuracy. The details are as follows.

*Figure 4.* **(A)** The distribution of distance between selected node pairs and the selected proportions of different pair distances in the pair gate. **(B)** The proportions of each output order being selected in the first Non-Diagonal block's tensor product gate. See the full cases of selected pairs in Appendix B.4.

**Three-phase Sparsity Scheduler.** We evaluated the effect of the three-phase sparsity scheduler on model performance by training SPHNet under various ablation settings: excluding the random stage, the adaptive stage, or the fixed stage. Each configuration was trained five times to assess performance stability and accuracy. As shown in Table 4, models trained without the random stage often converged to local optima, resulting in suboptimal performance. Models trained without the adaptive stage exhibited greater variance in outcomes and significantly lower accuracy compared to SPHNet with the full scheduler. Although omitting the fixed stage did not substantially affect accuracy, it reduced the speedup ratio to 5.45 due to the additional computational overhead from the sparse gate and the non-static graph implementation of tensor products following the Sparse Pair Gate, in contrast to the optimized implementations in the e3nn library. These findings highlight the importance of the three-phase sparsity scheduler in enhancing model performance and efficiency.

*Table 4.* Ablation Study on Three-phase Sparsity Scheduler.

| Random Stage | ✓ | ✗ | ✓ | ✓ |
|---|---|---|---|---|
| Adaptive Stage | ✓ | ✓ | ✗ | ✓ |
| Fixed Stage | ✓ | ✓ | ✓ | ✗ |
| $H$ [$10^{-6}E_h$] ↓ | 97.31 ±0.52 | 112.68 ±10.75 | 122.79 ±19.02 | 97.11 1.33 |
| Memory [GB/Sample] ↓ | 5.62 | 5.62 | 5.62 | 5.62 |
| Speed [Sample/Sec] ↑ | 3.12 | 3.12 | 3.12 | 3.12 |
| Speedup Ratio ↑ | 7.09× | 7.09× | 7.09× | 5.45× |

**Sparse Gates.** To further evaluate the impact of the Sparse Pair Gate and the Sparse Tensor Product Gate on overall performance, we trained SPHNet under three configurations: without the Sparse Pair Gate, without the Sparse Tensor Product Gate, and without both gates. As shown in Table 5, the Sparse Pair Gate achieved a 78% speedup and a 30% reduction in GPU memory usage, with minimal impact on model accuracy. The Sparse Tensor Product Gate, which removes 70% of the combinations in the tensor product, yielded a 160% speedup with an acceptable loss in accuracy. These results underscore the contribution of the Sparse Gate to improving model efficiency. Users may refer to Section 5.4 for guidance on choosing a sparsity ratio that balances speed and accuracy based on system size.

*Table 5.* Ablation Study on Sparse Gates.

| Sparse Pair Gate | ✓ | ✗ | ✓ | ✗ |
|---|---|---|---|---|
| Sparse TP Gate | ✓ | ✓ | ✗ | ✗ |
| $H$ [$10^{-6}E_h$] ↓ | 97.31 | 94.31 | 87.70 | 86.35 |
| Memory [GB/Sample] ↓ | 5.62 | 8.04 | 6.98 | 10.91 |
| Speed [Sample/Sec] ↑ | 3.12 | 1.75 | 1.20 | 0.76 |
| Speedup Ratio ↑ | 7.09× | 3.98× | 2.73× | 1.73× |

**Applying Sparse Gate to QHNet model.** To further assess the effectiveness of the Sparse Gates, we conducted additional experiments by integrating them into the QH-Net model. Specifically, the Sparse Pair Gate was applied to the second non-diagonal pair block, while the Sparse Tensor Product Gate was applied to the node-wise interaction blocks as well as both diagonal and non-diagonal pair blocks. As shown in the table below, the sparse gates significantly improved the training speed of the QHNet model and reduced computational resource usage. These results demonstrate the generalizability of sparse gating mechanisms across different SE(3)-equivariant networks.

*Table 6.* The effect of sparse gates on QHNet model.

| Sparse Pair Gate | ✗ | ✗ | ✓ | ✓ |
|---|---|---|---|---|
| Sparse TP Gate | ✗ | ✓ | ✗ | ✓ |
| $H$ [$10^{-6}E_h$] ↓ | 123.74 | 128.16 | 126.27 | 128.89 |
| Memory [GB/Sample] ↓ | 22.50 | 12.68 | 10.07 | 8.46 |
| Speed [Sample/Sec] ↑ | 0.44 | 0.90 | 0.73 | 1.45 |
| Speedup Ratio ↑ | 1.00× | 2.04× | 1.66× | 3.30× |

## 6. Conclusion and Future Work

In this work, we introduced SPHNet, an efficient and scalable SE(3) equivariant neural network for Hamiltonian prediction. By incorporating two adaptive sparse gates and a corresponding three-phase learning scheduler, SPHNet optimized both the number of tensor product operations and the efficiency of individual tensor computations. As a result, SPHNet achieved exceptional efficiency and performance in predicting Hamiltonians for larger molecular systems. Moreover, the proposed sparsification techniques demonstrated significant potential for extension to other SE(3) equivariant networks and broader prediction tasks.

This study opens several promising avenues for future work. First, the adaptive sparsity technique shows considerable potential for generalization to a wider range of tasks. Specifically, the sparse tensor product gates can be readily extended to any SE(3)-equivariant network architecture based on Clebsch-Gordan tensor products, while the sparse pair gates could be adapted to more types of pairwise interactions, such as those found in MACE (Batatia et al., 2022) for multi-body interactions. Second, integrating advanced loss functions, such as those introduced in (Li et al., 2025b), could further improve the accuracy of downstream properties of the Hamiltonian matrix, particularly in enhancing the applicability for large molecular systems. These directions will guide future efforts with further experimental validation, expanding the impact of adaptive sparsification and optimizing the performance of SE(3)-equivariant networks.

## Impact Statement

This paper presents work whose goal is to speed up the Hamiltonian matrix prediction process and advance the SE(3) equivariant network. There are many potential societal consequences of our work, none of which we feel must be specifically highlighted here.

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

# A. Additional Experimental Settings

## A.1. Training Setup.

The baseline model QHNet was using its default setting. The training setting for SPHNet is shown in Table 7, most training settings were set to align with the QHNet's experiment to make a fair comparison. The batch size for training is 10 on the MD17 dataset (except Uracil which was set to 5), 32 on the QH9 dataset, and 8 on the PubChemQH dataset. The training step was 260,000 for the QH9 dataset, 200,000 for the MD17 dataset, and 300,000 for the PubChemQH dataset. There was a 1,000-step warmup with the polynomial schedule. The maximum learning rate was 1e-3 for the QH9 dataset and PubChemQH dataset and was 5e-4 for the MD17 dataset. The training and test sets in the QH9 dataset were split in the same manner as their official implementation, including four different split sets. The MD17 dataset was randomly split into the train, validation, and test sets of the same size as the QHNet experiment. The PubChemQH datasets were split randomly into train, validation, and test sets by 80%, 10%, and 10%. The batch size for speed and GPU memory test was set to the maximum number that the GPU can hold to maximize its capability for fair comparison.

*Table 7.* Model Training Settings for Different Datasets.

| Dataset | Batch Size | Training Steps | Warmup Steps | Learning Rate | $L_{max}$ | Sparsity Rate | TSS Epoch $t$ | Train/Val/Test Split Method |
|---|---|---|---|---|---|---|---|---|
| MD17 Water | 10 | 200,000 | 1,000 | 5e-4 | 4 | 0.1 | 3 | Random (500/500/3900) |
| MD17 Ethanol | 10 | 200,000 | 1,000 | 5e-4 | 4 | 0.3 | 3 | Random (25000/500/4500) |
| MD17 Malondialdehyde | 10 | 200,000 | 1,000 | 5e-4 | 4 | 0.3 | 3 | Random (25000/500/1478) |
| MD17 Uracil | 5 | 200,000 | 1,000 | 5e-4 | 4 | 0.3 | 3 | Random (25000/500/4500) |
| QH9 | 32 | 260,000 | 1,000 | 1e-3 | 4 | 0.4 | 3 | Official (4 splits) |
| PubChemQH | 8 | 300,000 | 1,000 | 1e-3 | 6 | 0.7 | 3 | Random (40257/5032/5032) |

## A.2. Hardware and Software.

Our experiments are implemented based on PyTorch 2.1.0, PyTorch Geometric 2.5.0, and `e3nn` (Geiger et al., 2022) 0.5.1. In our experiments, the speed and GPU memory metrics are tested on a single NVIDIA RTX A6000 46GB GPU. The complete training of the models is carried on $4 \times$ 80GB Nvidia A100 GPU. Our code is available in supplementary material.

## A.3. Problem Formulation.

The loss function of SPHNet is the MAE plus MSE between predicted matrix $H_{pred}$ and ground truth matrix $H_{ref}$.

$$loss = MAE(\mathbf{H}_{ref}, \mathbf{H}_{pred}) + MSE(\mathbf{H}_{ref}, \mathbf{H}_{pred}). \tag{15}$$

SPHNet predicts the gap between the Hamiltonian matrix and the initial guess. The predicted target can be written as:

$$\Delta \mathbf{H} = \mathbf{H}_{ref} - \mathbf{H}_{init}, \tag{16}$$

where $\mathbf{H}_{init}$ is the initial guess of Hamiltonian matrix get with the `pyscf` (Sun et al., 2020) by function $init\_guess\_by\_minao(\cdot)$. On one hand, we observed that the scale and variance of $\Delta \mathbf{H}$ are approximately an order of magnitude smaller than those of $\mathbf{H}_{ref}$ on large datasets such as PubchemQH, which facilitates more stable learning across different datasets. On the other hand, the computational cost for calculating the initial guess is very cheap, it would not take long to obtain all the initial guesses of molecules in the dataset. Thus, we switched the training target to the $\Delta \mathbf{H}$ and added the $\mathbf{H}_{init}$ back to the predicted $\Delta \mathbf{H}$ to get the full Hamiltonian matrix.

## A.4. Evaluation Metric

**Mean Absolute Error (MAE) of Hamiltonian Matrix** $H$: This metric quantifies the Mean Absolute Error in relation to ground truth data obtained from Density Functional Theory (DFT), accounting for both diagonal and Non-Diagonal elements that reflect intra- and inter-atomic interactions, respectively. This can be presented as:

$$\mathbf{H} = mean(|\mathbf{H}_{pred} - \mathbf{H}_{gt}|), \tag{17}$$

where $H_{pred}$ is the predicted Hamiltonian matrix and $H_{gt}$ is the reference Hamiltonian matrix .

**Mean Absolute Error (MAE) of Occupied Orbital Energies** $\epsilon$: This metric evaluates the MAE of the occupied orbital energies, specifically the Highest Occupied Molecular Orbital (HOMO) and Lowest Unoccupied Molecular Orbital (LUMO)

levels. The accuracy of these critical properties is assessed by comparing the energies derived from the predicted Hamiltonian matrix against those obtained from the reference DFT calculations. The MAE can be expressed as:

$$\epsilon = \frac{1}{M} \sum_{k=1}^{M} |\epsilon_k^{\text{pred}} - \epsilon_k^{\text{ref}}|, \tag{18}$$

where $M$ represents the number of occupied orbitals, $\epsilon_k^{\text{pred}}$ and $\epsilon_k^{\text{ref}}$ are the predicted and reference energy of the k-th occupied orbital.

**Cosine Similarity of Orbital Coefficients** $\psi$: To evaluate the similarity between the predicted and reference electronic wavefunctions, we calculate the cosine similarity of the coefficients corresponding to the occupied molecular orbitals. This similarity metric is pivotal for understanding and predicting the chemical properties of the system. The cosine similarity $S$ can be defined as:

$$S(\psi^{\text{pred}}, \psi^{\text{ref}}) = \frac{\sum_i \psi_i^{\text{pred}} \psi_i^{\text{ref}}}{\|\psi^{\text{pred}}\| \|\psi^{\text{ref}}\|}, \tag{19}$$

where $\|\psi\|$ denotes the norm of the vector $\psi$, $\psi_i^{\text{ref}}$ and $\psi^{\text{ref}}$ are the Coefficient of the i-th occupied molecular orbital in the predicted and reference wavefunction

## B. Additional Experimental Results

### B.1. Ablation Study for blocks of SPHNet

We conducted an ablation study to evaluate the effect of different modules in the SPHNet architecture. Specifically, the standard SPHNet model has 4 Vectorial Node Interaction blocks, 2 Spherical Node Interaction blocks, and 2 Pair Construction blocks. We removed all the sparse gates and reduced the number of these three kinds of modules to 1, respectively, and observed the model performance. As shown in the table below, we found that both the Vectorial Node Interaction block and the Spherical Node Interaction block significantly affect the model performance, indicating that the design of architectures with progressively increased irreps orders has an important positive impact on the models. Interestingly, we found that removing one Pair Construction block would not strongly affect the model accuracy, suggesting that there is actually room to further speed up the model. We will explore this further in our future work.

*Table 8.* The effect of blocks' numbers in SPHNet on the PubChemQH dataset.

| Model | Sparse Pair Gate | Sparse TP Gate | Vectorial Node Interaction block | Spherical Node Interaction block | Pair Construction block | $H \downarrow$ [$10^{-6} E_h$] |
|---|---|---|---|---|---|---|
| SPHNet | ✗ | ✗ | 4 | 2 | 2 | 86.35 |
| SPHNet | ✗ | ✗ | 1 | 2 | 2 | 96.01 |
| SPHNet | ✗ | ✗ | 4 | 1 | 2 | 97.35 |
| SPHNet | ✗ | ✗ | 4 | 2 | 1 | 89.17 |

## B.2. The Effect of Sparity Rate on the Computational Cost

We analyze the effect of sparsity rate on the model performance in Section 5.4. Here we put the complete results of training speed and GPU memory in the PubChemQH dataset under every sparse rate. The results showed that the computational cost reduced linearly as the sparsity increased, which was in line with our expectations.

*Table 9.* The effect of sparsity rate v.s. computational cost on PubChemQH dataset

| Sparsity | H $[10^{-6}E_h]$ | Speed [Sample/Sec] | Memory [GB/Sample] |
|---|---|---|---|
| 0 | 78.48 | 0.78 | 17.02 |
| 10 | 79.90 | 0.96 | 12.49 |
| 20 | 81.29 | 1.05 | 11.21 |
| 30 | 85.33 | 1.24 | 10.93 |
| 40 | 86.65 | 1.50 | 6.84 |
| 50 | 90.52 | 1.86 | 6.22 |
| 60 | 91.90 | 2.31 | 5.12 |
| **70** | **97.31** | **3.12** | **5.62** |
| 80 | 158.70 | 3.92 | 4.68 |
| 90 | 183.52 | 5.13 | 3.38 |

## B.3. Single Tensor Product Time Scaling with Sparsity Rate

We test the time consumption for a single tensor product under different sparsity rates. As shown in Fig.6, the time consumption decreases linearly with increasing sparsity, in line with our expectations. This experiment suggested that the sparsity strategy can accelerate the tensor product speed on a linear scale.

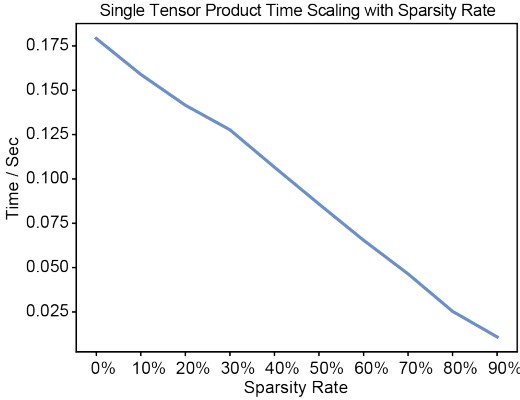

*Figure 6.* The time consumption for a single tensor product with scaling sparsity rates.

## B.4. Selected Combination Set Details

Here we presented the details of the selected combination sets in all Sparse TP Gate in the SPHNet. We first analysis the output order of the selected combinations. As shown in Fig.7, the selected combinations' output order in the Non-Diagonal block shows a monotonically decreasing trend as analyzed in section 5.5, suggesting the weakening of a single combination as the number of combinations within the same order increase. In the Diagonal block, we didn't observe any apparent pattern, this might be because all the combinations have similar effects on the model performance. However, this requires further analysis in the future.

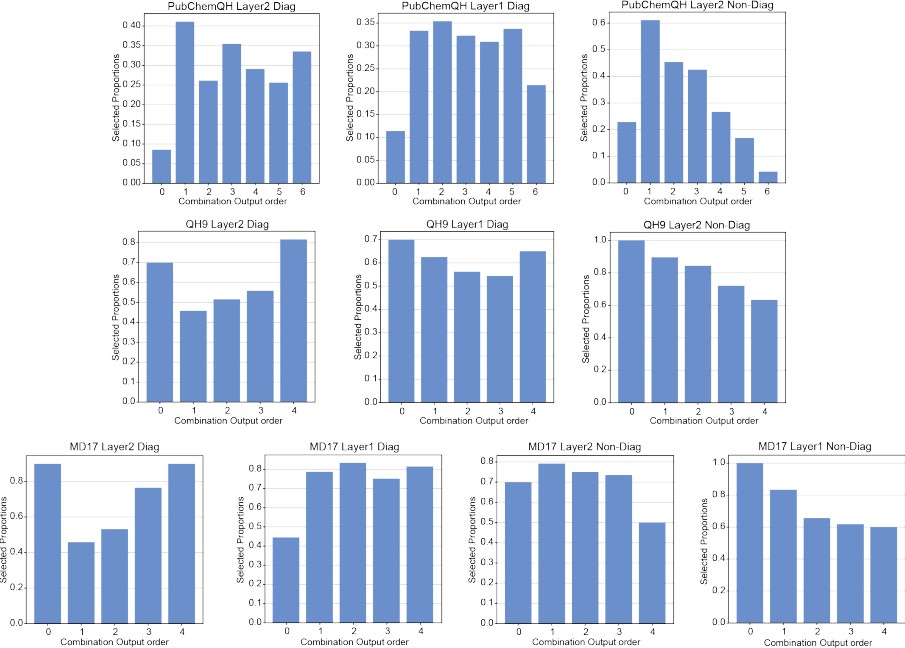

*Figure 7.* The proportions of each output order being selected in different tensor product gates.

We then check the combination weights for each of the selected combinations. The details are shown in Table 10, Table 11 and Table 12. These results are from the Sparse Tensor Product Gate before the first Non-Diagonal block. For the PubchemQH dataset, there are a total of 175 combinations in the tensor product (maximum irreps order 6), and the Sparse Pair Gate selects 30% of these combinations. For the QH9 dataset and MD17 dataset, there are a total of 65 combinations in the tensor product (maximum irreps order 4), and the Sparse Pair Gate selects 60% of these combinations for QH9, 70% of these combinations for MD17.

*Table 10.* Weight of selected combinations in PubChemQH

| Rank | Comb | Weight | Rank | Comb | Weight | Rank | Comb | Weight |
|------|------|--------|------|------|--------|------|------|--------|
| 1 | (4, 2, 3) | 1.1422 | 19 | (2, 5, 5) | 1.0736 | 37 | (4, 4, 2) | 1.0488 |
| 2 | (5, 4, 2) | 1.1398 | 20 | (3, 0, 3) | 1.0725 | 38 | (3, 1, 2) | 1.0530 |
| 3 | (0, 4, 4) | 1.1160 | 21 | (4, 3, 5) | 1.0710 | 39 | (4, 4, 2) | 1.0488 |
| 4 | (0, 3, 3) | 1.1093 | 22 | (0, 5, 5) | 1.0710 | 40 | (5, 2, 3) | 1.0486 |
| 5 | (3, 5, 2) | 1.1081 | 23 | (2, 3, 3) | 1.0682 | 41 | (2, 3, 1) | 1.0485 |
| 6 | (6, 6, 6) | 1.1078 | 24 | (4, 6, 2) | 1.0664 | 42 | (6, 5, 3) | 1.0482 |
| 7 | (5, 2, 5) | 1.1031 | 25 | (4, 6, 4) | 1.0651 | 43 | (3, 3, 4) | 1.0478 |
| 8 | (6, 3, 5) | 1.1028 | 26 | (6, 3, 3) | 1.0647 | 44 | (4, 5, 1) | 1.0464 |
| 9 | (4, 0, 4) | 1.1015 | 27 | (6, 5, 1) | 1.0607 | 45 | (5, 4, 1) | 1.0452 |
| 10 | (1, 4, 3) | 1.0978 | 28 | (3, 5, 4) | 1.0600 | 46 | (5, 0, 5) | 1.0417 |
| 11 | (2, 5, 3) | 1.0917 | 29 | (6, 4, 2) | 1.0596 | 47 | (3, 4, 1) | 1.0377 |
| 12 | (5, 3, 2) | 1.0903 | 30 | (2, 0, 2) | 1.0579 | 48 | (4, 3, 1) | 1.0300 |
| 13 | (2, 1, 3) | 1.0856 | 31 | (3, 2, 1) | 1.0566 | 49 | (1, 0, 1) | 1.0283 |
| 14 | (5, 5, 4) | 1.0854 | 32 | (5, 4, 4) | 1.0545 | 50 | (6, 6, 4) | 1.0281 |
| 15 | (4, 4, 4) | 1.0842 | 33 | (5, 6, 1) | 1.0537 | 51 | (5, 3, 4) | 1.0274 |
| 16 | (2, 4, 2) | 1.0835 | 34 | (4, 3, 3) | 1.0535 | 52 | (5, 5, 5) | 1.0248 |
| 17 | (3, 6, 3) | 1.0829 | 35 | (4, 5, 3) | 1.0534 | 53 | (2, 2, 2) | 0.9987 |
| 18 | (1, 2, 1) | 1.0799 | 36 | (3, 1, 2) | 1.0530 | | | |

*Table 11.* Weight of selected combinations in QH9

| Rank | Comb | Weight | Rank | Comb | Weight | Rank | Comb | Weight |
|------|------|--------|------|------|--------|------|------|--------|
| 1 | (1, 3, 2) | 1.1748 | 14 | (3, 1, 2) | 1.0758 | 27 | (3, 0, 3) | 1.0154 |
| 2 | (2, 4, 2) | 1.1722 | 15 | (2, 0, 2) | 1.0715 | 28 | (2, 3, 4) | 1.0150 |
| 3 | (3, 2, 1) | 1.1661 | 16 | (2, 2, 0) | 1.0696 | 29 | (4, 4, 1) | 1.0145 |
| 4 | (3, 4, 1) | 1.1433 | 17 | (3, 2, 3) | 1.0577 | 30 | (1, 0, 1) | 1.0145 |
| 5 | (3, 4, 3) | 1.1284 | 18 | (2, 3, 1) | 1.0530 | 31 | (0, 2, 2) | 1.0133 |
| 6 | (2, 1, 3) | 1.1240 | 19 | (0, 0, 0) | 1.0406 | 32 | (3, 4, 2) | 1.0117 |
| 7 | (3, 3, 2) | 1.1233 | 20 | (4, 4, 0) | 1.0383 | 33 | (1, 3, 3) | 1.0108 |
| 8 | (2, 2, 2) | 1.1163 | 21 | (4, 3, 1) | 1.0336 | 34 | (2, 2, 1) | 1.0099 |
| 9 | (1, 1, 0) | 1.1157 | 22 | (1, 2, 1) | 1.0319 | 35 | (4, 2, 3) | 1.0097 |
| 10 | (4, 4, 2) | 1.1085 | 23 | (1, 1, 1) | 1.0276 | 36 | (2, 3, 2) | 1.0090 |
| 11 | (0, 3, 3) | 1.0972 | 24 | (3, 4, 4) | 1.0213 | 37 | (3, 2, 2) | 1.0059 |
| 12 | (3, 3, 1) | 1.0853 | 25 | (0, 4, 4) | 1.0198 | 38 | (3, 1, 4) | 1.0057 |
| 13 | (0, 1, 1) | 1.0788 | 26 | (3, 3, 0) | 1.0154 | 39 | (4, 0, 4) | 0.9912 |

*Table 12.* Weight of selected combinations in MD17 (Ethanol)

| Rank | Comb | Weight | Rank | Comb | Weight | Rank | Comb | Weight |
|------|------|--------|------|------|--------|------|------|--------|
| 1 | (3, 2, 1) | 1.0755 | 16 | (3, 3, 1) | 1.0213 | 31 | (3, 3, 0) | 0.9894 |
| 2 | (3, 4, 1) | 1.0674 | 17 | (4, 4, 2) | 1.0206 | 32 | (4, 4, 1) | 0.9891 |
| 3 | (2, 1, 1) | 1.0608 | 18 | (2, 2, 1) | 1.0141 | 33 | (4, 1, 3) | 0.9886 |
| 4 | (1, 2, 1) | 1.0592 | 19 | (3, 1, 2) | 1.0119 | 34 | (2, 2, 2) | 0.9878 |
| 5 | (4, 3, 1) | 1.0449 | 20 | (2, 0, 2) | 1.0090 | 35 | (3, 2, 2) | 0.9856 |
| 6 | (0, 2, 2) | 1.0447 | 21 | (4, 4, 0) | 1.0037 | 36 | (4, 3, 3) | 0.9851 |
| 7 | (0, 3, 3) | 1.0446 | 22 | (2, 1, 3) | 1.0016 | 37 | (3, 3, 3) | 0.9811 |
| 8 | (2, 4, 2) | 1.0418 | 23 | (3, 3, 2) | 1.0003 | 38 | (3, 4, 4) | 0.9802 |
| 9 | (2, 3, 1) | 1.0417 | 24 | (3, 4, 2) | 0.9995 | 39 | (3, 4, 3) | 0.9795 |
| 10 | (1, 3, 2) | 1.0408 | 25 | (0, 4, 4) | 0.9991 | 40 | (1, 3, 4) | 0.9787 |
| 11 | (0, 1, 1) | 1.0399 | 26 | (3, 0, 3) | 0.9978 | 41 | (2, 3, 2) | 0.9769 |
| 12 | (1, 1, 2) | 1.0349 | 27 | (4, 3, 2) | 0.9972 | 42 | (3, 1, 3) | 0.9763 |
| 13 | (1, 0, 1) | 1.0330 | 28 | (3, 2, 3) | 0.9928 | 43 | (2, 4, 3) | 0.9736 |
| 14 | (4, 2, 2) | 1.0301 | 29 | (2, 3, 4) | 0.9920 | 44 | (2, 3, 3) | 0.9730 |
| 15 | (1, 4, 3) | 1.0295 | 30 | (0, 0, 0) | 0.9911 | 45 | (1, 2, 2) | 0.9719 |

### B.5. GPU Memory when Scaling with Increasing Size

We used molecules with different sizes to test the GPU memory consumption of SPHNet and the baseline model QHNet. As shown in Fig.8, we observed that the GPU memory consumption of SPHNet increased much slower than the baseline model. When the molecule had more than 1800 atomic orbitals, the baseline model reached the maximum GPU memory, which is 6 times the memory SPHNet needs, while the SPHNet can handle molecules with around 2900 atomic orbitals, making it possible to train on larger molecule systems.

## C. Additional Preliminary

### C.1. Spherical Harmonic Function

The spherical harmonic function in SPHNet is used to project the atomic vector $\vec{r_{ij}}$ into spherical space. A real basis for spherical harmonics $Y_{lm} : S^2 \to \mathbb{R}$ can be expressed in relation to their complex analogues $Y_m^l : S^2 \to \mathbb{C}$. This relationship is formulated through the following equations:

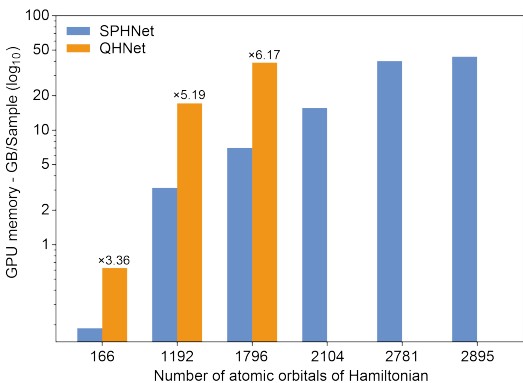

*Figure 8.* The GPU memory consumption with scaling molecule size.

$$Y_m^l = \begin{cases} \frac{i}{\sqrt{2}} Y_{-|m|}^l - (-1)^m Y_{|m|}^l, & \text{if } m < 0. \\ Y_0^l, & \text{if } m = 0. \\ \frac{1}{\sqrt{2}} \left( Y_{|m|}^l + (-1)^m Y_{-|m|}^l \right), & \text{if } m > 0. \end{cases} \tag{20}$$

To ensure both consistency and standardized treatment throughout the analysis, the Condon–Shortley phase convention is frequently utilized. The inverse relationships that define the complex spherical harmonics $Y_m^l : S^2 \to \mathbb{C}$ in terms of the real spherical harmonics $Y_{lm} : S^2 \to \mathbb{R}$ are outlined as follows:

$$Y_m^l = \begin{cases} \frac{1}{\sqrt{2}} \left( Y_{|m|}^{|l|} - i Y_{l,-|m|} \right), & \text{if } m < 0. \\ Y_{l,0}, & \text{if } m = 0. \\ (-1)^m \frac{1}{\sqrt{w}} \left( Y_{l,|m|} + i Y_{l,-|m|} \right), & \text{if } m > 0. \end{cases} \tag{21}$$

Established theories regarding the analytical solutions of the hydrogen atom indicate that the eigenfunctions corresponding to the angular component of the wave function are represented as spherical harmonics. Notably, in the absence of magnetic interactions, the solutions to the non-relativistic Schrödinger equation can also be represented as real functions. This aspect highlights the common use of real-valued basis functions in electronic structure calculations, as it simplifies software implementations by eliminating the need for complex algebra. It is crucial to acknowledge that these real-valued functions occupy a functional space identical to that of their complex counterparts, thus preserving generality and completeness in the solutions.

### C.2. Irreps and Tensor Products

**Irreps Representation**: SPHNet employs the special orthogonal group SO(3) to capture the essential 3D rotational symmetries intrinsic to molecular structures. It leverages the irreducible representations (irreps) of SO(3), indexed by an integer $l$, which are associated with spherical harmonic functions $Y_{lm}$. These spherical harmonics impart rotational characteristics to the feature vectors, thereby ensuring that the model maintains rotational invariance and facilitates consistent assessment of geometric properties.

**Tensor Product**: Tensor product is the core operation in SPHNet. It facilitates interactions among irreps linked to distinct order (angular momentum) $l$ and enhances expressiveness within the model. This operation merges two irreps with order $l_1$ and $l_2$ to generate a new irrep characterized by order $l_3$. The expansion is accomplished by utilizing Clebsch-Gordan coefficients, weighted by $w_{m_1,m_2}$:

$$(x^{\ell_1} \otimes y^{\ell_2})_{m_3}^{\ell_3} = \sum_{m_1=-\ell_1}^{\ell_1} \sum_{m_2=-\ell_2}^{\ell_2} w_{m_1,m_2}^{\ell_1,\ell_2,\ell_3} C_{(\ell_1,m_1),(\ell_2,m_2)}^{(\ell_3,m_3)} x_{m_1}^{\ell_1} y_{m_2}^{\ell_2}, \tag{22}$$

This mathematical approach allows for the amalgamation of complex features derived from simpler ones, effectively capturing the nuanced interactions of angular momentum in the molecular context.

## D. Additional Model Details.

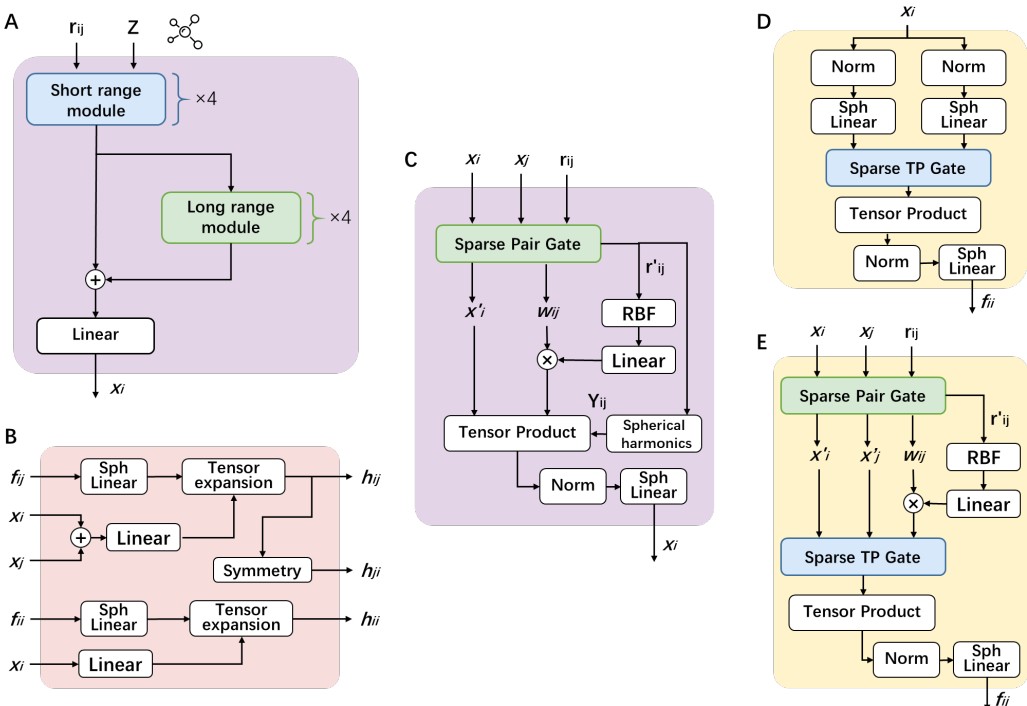

*Figure 9.* The components of SPHNet. **(A)** The Vectorial Node Interaction Block, which uses a long-short range message-passing mechanism. **(B)** The Expansion block. **(C)** The Spherical Node Interaction Block. **(D)** The Diagonal block in the Pair Construction block. **(E)** The Non-Diagonal block in the Pair Construction block.

### D.1. RBF

The RBF here refers to the Radial Basis Functions and is used to construct nonlinear maps of vectors $r$. We used Exponential Bernstein Radial Basis Functions here to process the vector of two atoms, and the mathematical formulation is as below:

$$\text{RBF}(r) = \text{F}_{cut}(r, \text{cutoff}) \times \exp\left(\log c + n \cdot (-\alpha r) + v \cdot \log(-\exp(-\alpha r) + 1)\right), \tag{23}$$

and the cutoff function $\text{F}_{cut}$ is to limit the range of action of the RBF, beyond which inputs will be ignored or reduced in impact:

$$\text{F}_{cut}(x, \text{cutoff}) = \begin{cases} \exp\left(-\frac{x^2}{(\text{cutoff}-x)(\text{cutoff}+x)}\right), & \text{if } x < \text{cutoff.} \\ 0, & \text{otherwise.} \end{cases} \tag{24}$$

### D.2. Sph Linear

The Sph Linear used in the SPHNet was implemented through the o3.Linear function in the `e3nn` (Geiger et al., 2022). It is an O(3) linear operation that takes an irrep as input and outputs the linear combination of input. Specifically, the feature of each order in the input irreps will be mapped to the feature of a specific order of the output irreps through a learnable matrix. We didn't set the instruction attribution in the `e3nn`'s (Geiger et al., 2022) function, which resulted in a fully connected instruction and each order in the input and output irreps will have a connection.

## D.3. Vectorial Node Interaction Block

As shown in Fig.9(A), the Node Vectorial Node Interaction Block receives the atomic numbers $Z$ and 3D coordinates of the molecular system $\vec{r}_{ij}$ as input and extracts the representations $\mathbf{x}_i$ for node $i$. In this block, we adopted the long-short range message-passing mechanism (Li et al., 2024) to capture the long-range interaction between each atom and obtain more informative node representations. There are 4 short-range message-passing modules and 4 long-range message-passing modules. Specifically, for the short-range message-passing module, the module passes the neighboring atoms' features to the center atom as defined below:

$$\hat{\mathbf{x}}_i^\ell = F_{short}(\mathbf{x}_i^\ell + \sum_{j\in\mathcal{N}(i)} \mathbf{m}_{ij}^\ell), \tag{25}$$

where $F_{short}$ is the linear layer, $\mathbf{m}_{ij}$ is the message from atom $j$ to atom $i$, and the $\mathcal{N}(i)$ is the neighboring atoms of atom $i$. For long-range message-passing module, the module passes the neighboring groups' feature to the center atom as defined below:

$$\hat{\mathbf{x}}_i^\ell = F_{long}(\mathbf{x}_i^\ell + \sum_{j\in\mathcal{N}(i)} \mathbf{m}_{ij}^\ell), \tag{26}$$

where $F_{long}$ is the linear layer, $\mathcal{N}(i)$ is the neighborhood group of atom $i$ on the atom-fragment bipartite graph, and $\mathbf{m}_{ij}$ is the message from neighboring group $j$ to atom $i$.

## D.4. Spherical Node Interaction Block

As shown in Fig.9(C), the Spherical Node Interaction Block takes the node feature $\mathbf{x}_i$ and 3D coordinates of the molecular system $\vec{r}_{ij}$ as input and outputs the new high-order feature through a tensor product operation. Specifically, the input atom feature $\mathbf{x}_i$ and its neighboring atom feature are first fed to the Sparse Pair Gate and obtain the selected important pair set $(\mathbf{x}_i', (\vec{r}_{ij})')$ and their pair weight $\mathbf{W}_p^{ij}$. This weight feature is then multiplied with the RBF of $\vec{r}_{ij}$ and gets the final tensor product weight as defined in Equation 9 and 10. The atom feature $x_i$ is finally doing tensor product with the high-order spherical harmonics projection of the selected $(\vec{r}_{ij})'$, as defined in Equation 12. Last, the ascended node feature $\hat{x}_i$ is outputted after normalization and sph linear:

$$\mathbf{x}_i^l = \begin{cases} \text{Layernorm}(\mathbf{x}_i^l), & \text{if } l = 0. \\ \mathbf{x}_i^l \times \frac{1}{L_{\max}} \sum_{l=1}^{L_{\max}} \left( \frac{1}{2l+1} \sum_{m=-l}^{l} {\mathbf{x}_{i,m}^l}^2 \right), & \text{if } l > 0. \end{cases} \tag{27}$$

$$\mathbf{x}_i = \text{Sphlinear}(\mathbf{x}_i). \tag{28}$$

There are two Spherical Node Interaction Blocks in the SPHNet. The first block increases the maximum order of input irreps from zero to the required order without utilizing the pairs selected by the Sparse Pair Gate. The second block inputs and outputs irreps with the same maximum order, and employs the pairs $(\mathbf{x}_i', (\vec{r}_{ij})')$ selected by the Sparse Pair Gate.

## D.5. Pair Construction Block

The Hamiltonian matrix contains the relational information of each pair of atoms in the molecular system. The objective of the Pair Construction Block is to extend the model's capacity to consider diagonal node pairs $\mathbf{f}_{ii}$ and non-diagonal node pairs $\mathbf{f}_{ij}$ in the Hamiltonian matrix. Therefore, there are two subblocks in the Pair Construction block, the Diagonal block for the diagonal feature and the Non-Diagonal block for the Non-Diagonal feature. There are two Pair Construction blocks in the SPHNet, to ensure that each subblock $h_{ij}$ in the Hamiltonian matrix is covered at least once, the first Pair Construction block does not use the pair selected by Sparse Pair Gate.

### D.5.1. DIAGONAL BLOCK

As shown in Fig.9D, the Diagonal block first uses two separate sph linear layers to get two different representations from the input node feature, and then uses a self-tensor product operation to obtain $\mathbf{f}_{ii}$. Here, the combinations in the tensor product are selected by the Sparse Tensor Product Gate:

$$\hat{\mathbf{x}}_{i_1} = F_{s_1}(\hat{\mathbf{x}}_i), \hat{\mathbf{x}}_{i_2} = F_{s_2}(\hat{\mathbf{x}}_i), \tag{29}$$

$$\mathbf{f}_{ii}^{\ell_3} = \sum_{\{\ell_1,\ell_2,\ell_3\}\in U_c^{\text{TSS}}} \mathbf{W}_{ii}^{(\ell_1,\ell_2,\ell_3)} \times (\hat{\mathbf{x}}_{i_1}^{\ell_1} \otimes_{\ell_1,\ell_2}^{\ell_3} \hat{\mathbf{x}}_{i_2}^{\ell_2}), \tag{30}$$

where $\mathbf{x}_i$ is the atom feature from the Spherical Node Interaction Block, and $F_{s_1}$ and $F_{s_2}$ are the sph linear layer. $\mathbf{W}_{ii}$ is the combination weight defined in Equation 13. Last, the diagonal feature $\mathbf{f}_{ii}$ is outputted after normalization and sph linear the same as in the Spherical Node Interaction block.

### D.5.2. Non-Diagonal Block

For the non-diagonal pairs, the module first uses Sparse Pair Gate to select the most valuable pair $\{\mathbf{x}'_i, \mathbf{x}'_j\}$ and their tensor product weight $\mathbf{w}_{ij}$, as defined in Equation 9. The tensor product weight is then multiplied with the RBF of $\vec{r}_{ij}$ and gets the final tensor product weight as defined in Equation 12. Then, the module calculates the tensor product of $\mathbf{x}'_i$ and $\mathbf{x}'_j$, the process is presented in Fig.9E. Note that the combinations in the tensor product are selected by the Sparse Tensor Product Gate. The result is regarded as the desired $\mathbf{f}_{ij}$, defined in Equation 14. Last, the non-diagonal feature $\mathbf{f}_{ij}$ is outputted after normalization and sph linear the same as in the Spherical Node Interaction block.

There are two separate Pair Construction Blocks that receive atom representations from the two SO(3) Convolution Blocks respectively, and the final node pair feature is the addition of these two Pair Construction Blocks' output.

### D.6. Expansion Block

Once we have gathered the irreps for both diagonal and non-diagonal pairs, the subsequent step involves constructing the complete Hamiltonian matrix. There are many matrix elements in the Hamiltonian matrix, each matrix element represents the interaction between two orbitals. Specifically, the block denoted as $h_{ij}$ captures all interactions between atoms $i$ and $j$. Since atoms have varying numbers of orbitals, the shape of the pair block $h_{ij}$ is different, and must be determined during the construction of the Hamiltonian matrix.

Here, same as the previous work (Yu et al., 2023), we introduce an intermediate block $M_{ij}$ containing the full orbitals, from which we can derive $h_{ij}$ by extracting the relevant components based on the specific atom types. For instance, in the MD17 dataset, there are four atoms—H, C, N, and O—each with its own set of orbitals. The full orbital set consists of 1s, 2s, 3s, 2p, 3p, 3d orbitals, where $M \in \mathbb{R}^{14\times14}$. When dealing with the hydrogen atom (H), only the 1s, 2s, and 2p orbitals are selected to contribute to the Hamiltonian matrix. By using this strategy, each node's irrep $\mathbf{f}_{ij}$ can be converted into an intermediate block $M_{ij}$ with a predetermined structure, irrespective of the atom type. This technique is particularly advantageous as it can be easily adapted to various molecules.

To construct the intermediate pair blocks with full orbitals using pair irreducible representations, we apply a tensor expansion operation in conjunction with the filter operation. This expansion is defined by the following relation:

$$\left(\overline{\otimes}_{\ell_o}\mathbf{f}^{\ell_o}\right)_{(\ell_i,\ell_j)}^{(m_i,m_j)} = \sum_{m_o=-\ell_o}^{\ell_o} C_{(\ell_i,m_i),(\ell_j,m_j)}^{(\ell_o,m_o)}\mathbf{f}_{m_o}^{\ell_o}, \tag{31}$$

where $C$ denotes the Clebsch-Gordan coefficients, and $\overline{\otimes}$ symbolizes the tensor expansion which is the converse operation of the tensor product. Specifically, $x^{\ell_i} \otimes y^{\ell_j}$ can be expressed as a sum over tensor expansions:

$$\mathbf{x}^{\ell_i} \otimes \mathbf{y}^{\ell_j} = \sum_{\ell_3} \mathbf{W}_{l_i,l_j,l_o}\overline{\otimes}\mathbf{f}^{\ell_o}, \tag{32}$$

subject to the angular momentum coupling constraints $|\ell_i - \ell_j| \leq \ell_o \leq \ell_i + \ell_j$. The filter then takes the atom types as input, producing a weight for each path $(\ell_{o1}, \ell_{o2}, \ell_{in})$:

$$\mathbf{F}_{ij}^{(\ell_{o1},\ell_{o2},\ell_{in})} = f(Z_i, Z_j), \quad \mathbf{F}_{ii}^{(\ell_{o1},\ell_{o2},\ell_{in})} = f(Z_i), \tag{20}$$

where $Z$ denotes the embedding of the atom types. The intermediate blocks $M$ are generated by the filter, using the node pair irreducible representations as follows:

$$M_{ii}^{(\ell_{o1},\ell_{o2})} = \sum_{\ell_{in},c'} \mathbf{F}_{ii}^{(\ell_{o1},\ell_{o2},\ell_{in})}\overline{\otimes}\mathbf{f}_{ii}^{(\ell_{in},c')}, \tag{21}$$

$$M_{ij}^{(\ell_{o1},\ell_{o2})} = \sum_{\ell_{in},c'} \mathbf{F}_{ij}^{(\ell_{o1},\ell_{o2},\ell_{in})} \overline{\otimes} \mathbf{f}_{ij}^{(\ell_{in},c')}. \tag{33}$$

Here, $c'$ indicates the channel index in the input irreducible representations, and $c$ represents the channel index in $M$. For instance, there are nine channels with $(\ell_1,\ell_2) = (0,0)$ and four channels with $(\ell_1,\ell_2) = (1,1)$. A bias term is added for the node pair representation when $\ell_{in} = 0$.

Last, we derive $\mathbf{h}_{ij}$ by extracting the relevant components based on the specific atom types and use these matrix elements to construct the complete Hamiltonian matrix. As shown in Fig.9(B), for the Non-Diagonal block in the Hamiltonian matrix, we only construct the node pair features $\mathbf{f}_{ij}$ from the upper triangular part of the Hamiltonian matrix. The lower triangular part is then obtained by symmetrizing the sub-blocks from the upper triangular part.

### D.7. Computational Overhead of the Sparse Gates and Scheduler

Compared to the tensor product operation, Sparse Gate and the three-phase learning schedule only introduce minimal computational overhead and have little impact on the overall speed. We would like to discuss this part of computational cost here.

In the three-phase sparsity scheduler, for a given unsparsified set $U$, the additional computational overhead in the first phase has a complexity of $\mathcal{O}(|U|)$, contributed by the RANDOM($\cdot$) operation. The second phase has a computational overhead of $\mathcal{O}(|U|\log|U|)$, arising from the TOP($\cdot$) operation. Since we fix the learnable weight matrix and the selected elements, there is no additional computational overhead in the third phase. For detailed information, please refer to Equation 3.

For the sparse TP gate, the computational overhead comes from the element-wise multiplication of two weight vectors (Equation 5), so its complexity is $\mathcal{O}(|U_c|) = \mathcal{O}(L^3)$.

For the sparse pair gate, the additional computational overhead mainly comes from the linear layer $F_p(\cdot)$ in Equation 7, with its time complexity being the square of its hidden dimension. Other operations, including the inner product (Equation 6) and the weight calculation (Equation 9), are necessary operations in our framework, even without the sparse pairwise gate.

## E. Additional Related works

### E.1. SE(3) Equivariant Neural Network

The SE(3) equivariant neural network is one of the most used models in the field of AI for chemistry (Fuchs et al., 2020; Du et al., 2022; Musaelian et al., 2023; Liao & Smidt, 2022; Liao et al., 2023; Batzner et al., 2022; 2023). The fundamental feature of SE(3) equivariant neural network is that all the features and operations in the model are SE(3) equivariant. This is achieved by using irreducible representations (irreps) and functions of geometry built from spherical harmonics. In the network, the equivariant features propagate through each layer and interact with other features by equivariant operations, and finally obtain the desired SE(3) equivariant quantity.

SE(3)-Transformer (Fuchs et al., 2020) proposed a novel self-attention mechanism for 3D point clouds and graphs that ensures robustness through continuous 3D roto-translation equivariance, which is widely used in the field of quantum chemistry. The Equiformer (Liao & Smidt, 2022) introduced a novel graph neural network leveraging SE(3) Transformer architecture based on irreducible representations to predict molecule property. It demonstrated strong empirical results by incorporating tensor products and a new attention mechanism called equivariant graph attention. The EquiformerV2 (Liao et al., 2023) is an improved version of Equiformer, which scales effectively to higher-order representations by replacing SO(3) convolutions with efficient eSCN convolutions (Passaro & Zitnick, 2023), and outperforming the traditional network such as GemNet (Gasteiger et al., 2022) and Torchmd-Net (Thölke & De Fabritiis, 2022) in molecular energy and force prediction and other downstream tasks. Allegro (Musaelian et al., 2023) used a strictly local, equivariant model to represent many-body potential using iterated tensor products of learned representations without atom-centered message passing. It demonstrates remarkable generalization to out-of-distribution data and accurately recovers structural and kinetic properties of amorphous electrolytes in agreement with ab initio simulations.

These SE(3) equivariant neural networks greatly improve the performance of Artificial Intelligent in the field of quantum chemistry. However, the computational complexity of tensor product operation greatly reduces the efficiency of these SE(3) equivariant models. The eSCN (Passaro & Zitnick, 2023) presents an efficient method to perform SO(3) equivariant

convolutions. It reduces the computational complexity by aligning node embeddings' primary axis with edge vectors, transforming the SO(3) convolutions into mathematically equivalent SO(2) convolutions, which decreases complexity from $O(L^6)$ to $O(L^3)$. E2Former (Li et al., 2025a) introduces an equivariant and efficient transformer architecture that incorporates the Wigner 6j convolution (Wigner 6j Conv). By shifting the computational burden from edges to nodes, the Wigner 6j Conv reduces the complexity from $O(|E|)$ to $O(|V|)$ while preserving both the model's expressive power and rotational equivariance.

### E.2. Hamiltonian Matrix Prediction

Predicting the Hamiltonian matrix (also known as electronic wavefunctions) is gradually gaining more and more attention because of the wide range of potential application scenarios of the Hamiltonian matrix. There are more and more works trying to solve the problem of predicting the Hamiltonian matrix using deep learning techniques.

The SE(3) equivariant neural network is one of the most used models in the field of Hamiltonian matrix prediction (Unke et al., 2021; Yu et al., 2023; Gong et al., 2023; Atz et al., 2021). The PhiSNet (Unke et al., 2021) is the first method that primarily focuses on Hamiltonian matrix prediction. It leverages SE(3)-equivariant operations through its whole architecture to predict molecular wavefunctions and electronic densities, and can reconstruct wavefunctions with high accuracy. The main problem for PhiSNet is its inefficiency due to the large amount of tensor product operations. To solve this problem, QHNet (Yu et al., 2023) proposed a model with careful design to greatly reduce the number of tensor products and improve the efficiency of Hamiltonian prediction. The DeepH-E3 (Gong et al., 2023) is an E(3)-equivariant model that preserves Euclidean symmetry even with spin-orbit coupling. The method allows for efficient and accurate electronic structure calculations of large-scale materials by learning from small-sized DFT data, significantly reducing computational costs.

There are also other works that try to enhance the prediction ability from different aspects. DeepH (Li et al., 2022) introduces a local coordinate system defined for each edge according to its local chemical environment to handle the gauge (or rotation) covariance of the DFT Hamiltonian matrix. This allows the Hamiltonian matrix blocks to be invariant under rotation when transformed into the local coordinate system. Self-consistency training (Zhang et al., 2024) leverages the self-consistency principle of density functional theory (DFT) to train a model without requiring the labeled Hamiltonian matrice. This enhances the generalization and efficiency and reduces reliance on costly DFT calculations for supervision. The DEHQ (Wang et al., 2024) integrates Deep Equilibrium Models (DEQs) to predict Hamiltonians. It inherently captures the self-consistent nature of Hamiltonians, a critical aspect often overlooked by traditional machine learning methods. By employing DEQs, the model circumvents the need for iterative DFT calculations during training. WANet (Li et al., 2025b) introduces a scalable deep learning model for Hamiltonian prediction. By leveraging a novel Wavefunction Alignment Loss (WALoss), the model notably reduces total energy error derived from the predicted Hamiltonian matrix and accelerates SCF calculations with the predicted Hamiltonian matrix.

