# OpenReview forum: "Efficient and Scalable Density Functional Theory Hamiltonian Prediction through Adaptive Sparsity"
_ICML.cc/2025/Conference — ICML 2025 poster_

### Official Review · Reviewer_EBLX · 2025-03-10

**Overall Recommendation:** 4

**Summary:**

The paper presents a significant advancement in SE(3) equivariant neural networks by introducing a scalable and efficient approach for Hamiltonian prediction. Through innovative sparse gating mechanisms and an adaptive training scheduler, SPHNet achieves remarkable computational savings without sacrificing accuracy.

**Claims And Evidence:**

Yes

**Essential References Not Discussed:**

No

**Experimental Designs Or Analyses:**

Experiments follow previous baseline methods.

**Methods And Evaluation Criteria:**

Yes

**Other Comments Or Suggestions:**

N/A

**Other Strengths And Weaknesses:**

1. The paper addresses a critical bottleneck in SE(3) equivariant graph neural networks—high computational cost due to tensor product (TP) operations.

2. The Sparse Pair Gate filters out unimportant node pairs, reducing computational overhead. The Sparse TP Gate prunes insignificant interactions across different tensor product orders, improving efficiency while maintaining performance.

3. The proposed Three-phase Sparsity Scheduler enables stable training and convergence by progressively optimizing sparse representations, ensuring the balance between efficiency and precision.

4. Experimental results are good. But I'm curious about the effect of each module on the model performance. It would be better if authors can provide more experiments studying the contribution of each module.

**Questions For Authors:**

N/A

**Relation To Broader Scientific Literature:**

The Sparse TP and Pair Gates could be extended to other SE(3)-equivariant networks, benefiting a wide range of applications in computational chemistry. For example, molecular energy and force field predictions in addition to the Hamiltonian matrix prediction.

**Theoretical Claims:**

Yes

---

> ### Author Rebuttal · Authors · 2025-04-01
>
> We thank all your valuable comments very much, and would like to discuss more on the 4th question of ‘Other Strengths And Weaknesses’:
>
> We had conducted a series of ablation study to examine the contribution of each module, and the results had been included in the Appendix B.2. As shown in Appendix Table 6 (we also listed the results below), we removed the Sparse Pair gate, the Sparse TP gate, and the Vectorial Node Interaction block from the SPHNet model, respectively, to see their impact on the model efficiency. We found that the two kinds of sparse gates can improve the speedup ratio from 1.73x to 7.09x, demonstrating the effect of sparse gates on the model acceleration. Also, each of the Sparse pair gate and Sparse TP gate alone can provide significant acceleration, improving the speedup ratio from 3.98x to 7.09x and 2.73x to 7.09x. You can find the detailed analysis in Appendix B.2.
>
> Additionally, we add a similar study by adding the Sparse pair gate and Sparse TP gate to the QHNet model here. Specifically, we applied the Sparse Pair gate on the second Non-diagonal pair block and applied the Sparse TP gate on the Node-wise interaction blocks and both Diagonal and Non-diagonal pair block. As shown in the table below, the sparse gate can effectively accelerate the training speed of the QHNet model and save computational consumption. We will also add this part of the experiment to the Appendix in our future version also.
>
> **Table:** The effect of sparse gates on SPHNet/QHNet model on PubChemQH dataset.
>
> | Model | Sparse Pair Gate | Sparse TP Gate | Vectorial Node Interaction block | Spherical Node Interaction block | *H* [$10^{-6}E_h$] | Memory [GB/Sample] ↓ | Speed [Sample/Sec] ↑ | Speedup Ratio ↑ |
> |-------|--------------------|------------------|-------------------------------------|------------------------------------|-----------------------------|-------------------------|-------------------------|--------------------|
> | SPHNet | ✓ | ✓ | 4 | 2 |  97.31  | 5.62 | 3.12 | 7.09x |
> | QHNet | ✗ | ✗ | 0 | 5 |  123.74  | 22.50 | 0.44 | 1.00x |
> | SPHNet | ✗ | ✓ | 4 | 2 | 94.31  | 8.04 | 1.75 | 3.98x |
> | SPHNet | ✓ | ✗ | 4 | 2 | 87.70  | 6.98 | 1.20 | 2.73x |
> | SPHNet | ✗ | ✗ | 4 | 2 | 86.35  | 10.91 | 0.76 | 1.73x |
> | SPHNet | ✓ | ✓ | 0 | 5 | 97.08  | 8.47 | 1.08 | 2.45x |
> | QHNet | ✗ | ✓ | 0 | 5 | 128.16  | 12.68 | 0.90 | 2.04x |
> | QHNet | ✓ | ✗ | 0 | 5 | 126.27  | 10.07 | 0.73 | 1.66x |
> | QHNet | ✓ | ✓ | 0 | 5 | 128.89  | 8.46 | 1.45 | 3.30x |
>
>
> Besides, we conduct an extra ablation study to evaluate the effect of different modules in the SPHNet architecture. Specifically, the standard SPHNet model has 4 Vectorial Node Interaction blocks, 2 Spherical Node Interaction blocks, and 2 Pair Construction blocks. We removed all the sparse gates and reduced the number of these three kinds of modules to 1, respectively, and observed the model performance. As shown in the table below, we found that both Vectorial Node Interaction block and Spherical Node Interaction block significantly affect the model performance, indicating the design of architectures with progressively increased irreps orders has an important positive impact on the models. Interestingly, we found that remove one Pair Construction block would not strongly affect the model accuracy, suggesting that there is actually room to further speed up the model. We will explore this further in our future work.
>
>
> **Table:** The effect of different modules on SPHNet model on PubChemQH dataset.
>
> | Model  | Sparse Pair Gate | Sparse TP Gate | Vectorial Node Interaction block | Spherical Node Interaction block | Pair Construction block | *H* [$10^{-6}E_h$] ↓ |
> |--------|------------------|----------------|----------------------------------|----------------------------------|-------------------------|----------------------|
> | SPHNet | ✗                | ✗              | 4                                | 2                                | 2                       | 86.35                |
> | SPHNet | ✗                | ✗              | 1                                | 2                                | 2                       | 96.01                |
> | SPHNet | ✗                | ✗              | 4                                | 1                                | 2                       | 97.35                |
> | SPHNet | ✗                | ✗              | 4                                | 2                                | 1                       | 89.17                |

---

### Official Review · Reviewer_28vQ · 2025-03-14

**Overall Recommendation:** 3

**Summary:**

In this paper, the author proposes a new efficient equivariant operation based on Tensor Product (TP), named Sparse tensor product gate, to improve the efficiency of equivariant networks for Hamiltonian matrix prediction task. From the experiment, the proposed model achieves SOTA performance on QH9 and PubCHemQH, while improving the efficiency about 3-7 times faster than previous method.

**Claims And Evidence:**

The experiments are valid and well support the claim of this paper.

**Essential References Not Discussed:**

N/A.

**Experimental Designs Or Analyses:**

Sound evaluations and experiments.

**Methods And Evaluation Criteria:**

Sound evaluations. A comprehensive benchmark on existing datasets including QH9 and PubChemQH with reasonable metrics including the MAE on Hamiltonian matrix, MAE on eigen energies, and cosine similarity on the electronic wavefunction. The proposed method greatly improves the model efficiency, which is important for the training and inference of Hamiltonian matrix prediction task.

**Other Comments Or Suggestions:**

N/A.

**Other Strengths And Weaknesses:**

Strength:

1. The proposed technique is interesting and valid. From the experiments, the sparse adaptive TP can greatly improve the efficiency while the performance will not drop a lot through the ablation study in Appendix B.1. This well supports the main motivation of this paper.

2. The experimental results are strong, achieving the SOTA performance on QH9 and PubChemQH.

3. The writing and organization of this paper is clear.

Weakness:
1. From the experiments, although the performance improves a lot, the reasons for such improvement have not been well discussed. Since the sparse adaptive TP mainly aims to improve efficiency, I assume the usage of such operation will not bring performance improvement. Therefore, more ablation studies should be included to discuss this.

**Questions For Authors:**

Questions:

1. Would you mind sharing the source of the PubChemQH datasets? It seems they are not publicly available yet.

2. I find the time/sec is linearly decreased with the increasing of sparsity rate. How about the relationship of model performance with sparsity ratio?

**Relation To Broader Scientific Literature:**

N/A.

**Theoretical Claims:**

N/A.

---

> ### Author Rebuttal · Authors · 2025-03-31
>
> Thanks for your valuable comments and suggestions. We address each of the comments individually below.
>
> **Weakness：**
>
> Your assumption regarding the effect of the sparse gate is correct. As demonstrated in the ablation study presented in Table 6 of Appendix B.2, both types of sparse gates significantly improve speed with only a slight loss in accuracy. Therefore, compared to QHNet, the performance improvement in accuracy of our method primarily stems from SPHNet’s architectural design. Given that SPHNet has already achieved strong performance, the minor accuracy loss due to sparsity is entirely acceptable. For more details on the ablation study of different sparse gate, please refer to Appendix B.2 of the paper. Additionally, further experiments on applying the sparse gate to QHNet can be found in our response to the 4th Reviewer EBLX.
>
> Besides, to further address your concerns, we conducted an ablation study to evaluate the effect of different modules in the SPHNet architecture. Specifically, the standard SPHNet model has 4 Vectorial Node Interaction blocks, 2 Spherical Node Interaction blocks, and 2 Pair Construction blocks. We removed all the sparse gates and reduced the number of these three kinds of modules to 1, respectively, and observed the model performance. As shown in the table below, we found that both Vectorial Node Interaction block and Spherical Node Interaction block significantly affect the model performance, indicating the design of architectures with progressively increased irreps orders has an important positive impact on the models. Interestingly, we found that remove one Pair Construction block would not strongly affect the model accuracy, suggesting that there is actually room to further speed up the model. We will explore this further in our future work.
>
> **Table:** The effect of different modules on SPHNet model on PubChemQH dataset.
>
> | Model  | Sparse Pair Gate | Sparse TP Gate | Vectorial Node Interaction block | Spherical Node Interaction block | Pair Construction block | *H* [$10^{-6}E_h$] ↓ |
> |--------|------------------|----------------|----------------------------------|----------------------------------|-------------------------|----------------------|
> | SPHNet | ✗                | ✗              | 4                                | 2                                | 2                       | 86.35                |
> | SPHNet | ✗                | ✗              | 1                                | 2                                | 2                       | 96.01                |
> | SPHNet | ✗                | ✗              | 4                                | 1                                | 2                       | 97.35                |
> | SPHNet | ✗                | ✗              | 4                                | 2                                | 1                       | 89.17                |
>
> However, since our model has very different architecture and components from other models, and the specific module in the model is not a one-to-one correspondence, it's hard to make a substitution of the specific module and carry on the ablation study with the other model like QHNet. We are sorry that we are not able to carry on the ablation study with a different model and explain why the SPHNet has such performance improvements compared to others. We'd like to give a try in the future to explore more about this.
>
> **Questions For Authors 1:**
>
> Thank you very much for your interest! We are very glad to share our data with the whole community. But our organization require a review process for open sourcing data, which might take some time. We are already actively promoting this data open source process and hope to be able to share our data soon.
>
> **Questions For Authors 2:**
>
> Thank you for your question. We have actually evaluated the relationship between model performance and sparsity ratio in Section 5.4. As shown in Figure 3, for all three datasets, the Hamiltonian MAE remained stable within a certain sparsity range. However, when the sparsity rate reached a particular threshold, we observed a significant increase in the Hamiltonian MAE, which we interpret as the upper limit of sparsity. This suggests that a suitable range of sparsity has little impact on model accuracy while significantly improving computational efficiency. We provide a detailed analysis of this in Section 5.4.

---

### Official Review · Reviewer_hk4X · 2025-03-14

**Overall Recommendation:** 3

**Summary:**

This paper tackles the Hamiltonian prediction task. It proposes to learn a mask to select important pairs in pair-wise interactions in both node interactions and non-diagonal pair construction blocks. Moreover, the paper also use similar techniques to select important paths in the tensor product during pair construction. The proposed SPHNet can reduce the computational cost. The experiments are conducted on QH9, PubChemQH and MD17 datasets.

**Claims And Evidence:**

- The paper compares SPHNet and QHNet (as well as WANet) for speedup. However, the architecture of SPHNet and QHNet are not the same. As a result, the speed-up ratio may not entirely come from the sparse gates.

- Figure 3 shows the effect of sparsity rate on prediction accuracy, it would be interesting to see the effect of sparsity rate v.s. computational cost for training and testing.

**Essential References Not Discussed:**

Not I am aware of.

**Experimental Designs Or Analyses:**

- The result for water molecule in Table 3 does not seem to be good. Is it because of using the sparse gates?

**Methods And Evaluation Criteria:**

Yes.

**Other Comments Or Suggestions:**

- Running title is not formatted.

- Section 3: $C \in \mathbb{R}^{n\times n}$, is the dimension for coefficient correct?

**Other Strengths And Weaknesses:**

- In section 4.1: "TOP(·) is used to select the elements with the highest weights from the set with a given probability 1 − k", what does it mean? Is there still randomness? Can the path selection still get updated during training when using TOP?

- In equation 12, there is superscripts $\ell_1, \ell_2, \ell_3$ for the weights $w_{ij}$. However in equation 9 there is not. What is the shape of $w_{ij}$?

- The proposed sparsity selection introduces additional hyper parameters, including the scheduler step $t$ and the sparsity rate. The sparsity rate may negatively affect the performance if not chosen adequently.

**Questions For Authors:**

- Can the choices of retained paths be updated during the second phase of sparsity scheduler?

- How are the sparsity weights initialized?

- In equation 6, what does the superscript 0 of the inner product mean?

**Relation To Broader Scientific Literature:**

They are adequately discussed.

**Theoretical Claims:**

The equations mainly server to describe the method.

---

> ### Author Rebuttal · Authors · 2025-03-31
>
> Thank you very much for your valuable comments and suggestions. Below are our detailed responses.
>
> **Claims And Evidence**
>
> 1. Thank you for your question. As you noted, SPHNet’s lightweight design allows it to run 1.73× faster than QHNet on the PubChemQH dataset. However, the primary acceleration comes from sparse gates. To isolate their effect, we included ablation study in Appendix B.2. Beside, to further address your concerns, we test these gates on QHNet. Please refer to our answer to the 4th reviewer EBLX for the details due to the length limitation of the response.
>
> 2. The computational cost of key sparsity has already been included in Figure 3, where the number above each “$\times$” symbol represents the training speed at that specific sparsity level. In Appendix Figure 8, we provide a more detailed visualization of how time scales with sparsity. Additionally, we have listed the complete results for training speed and GPU memory usage at each sparsity level on the PubChemQH dataset in the [table](https://imgur.com/a/tM8STT4) below. The results indicate that as sparsity increases, both time and memory costs decrease in an approximately linear manner.
>
> **Experimental Designs Or Analyses:**
>
> 1. We agree with your opinion that the sparse gates cause suboptimal results. As we had analysis in Section 5.3, since the water molecule is very small (only 3 atoms) compared to other molecules, there is not much room for the atom pairs and TP combination reductions. Therefore, spare gates may remove the necessary interaction combinations within the system and cause poor results.
>
>     However, we would like to declare that the sparse gates are not designed for these extremely small molecules, and we are more concerned about their performance on larger molecules, which turn out to be very efficient on the large molecule dataset PubChemQH.
>
> **Other Strengths And Weaknesses**
>
> 1. Sorry for the confusion caused by the word "probability." A clearer term would be "percentage." Specifically, in the second phase, TOP(·) selects elements whose weights are within the top $( 1 - k )$ percent, with no randomness involved—selection is purely based on learnable weights. We have revised the manuscript for clarity.
>
>     Regarding path selection, the learnable weights continue to update in phase two, as they participate in subsequent operations (Equations 4 and 5). Only selected paths' weights are updated, while TOP(·)  consistently selects the highest-weighted elements. As weights evolve, the selected paths adjust accordingly, allowing the sparse gate to gradually learn the optimal path set.
>
> 2. Thank you for the question. The $w_{ij}$ is a $R^{k}$ vector, where $k$ is the number of elements in the complete set $ U_c= \{(\ell_1, \ell_2, \ell_3) \mid \ell_3 \in [|\ell_1 - \ell_2|, \ell_1 + \ell_2] \} $ in the tensor product. And the $w_{ij}^{\ell_1, \ell_2, \ell_3}$ is a single number from the $w_{ij}$ vector. Note that in Equation 12 we selected the $ w^{\ell_1, \ell_2, \ell_3}_{ij} $ that within the $U^{TSS}_p$ set.
>
> 3. We used a fixed scheduler step $( t = 3 )$ across all datasets and experiments, as results remained stable, indicating minimal impact on performance. We recommend setting $( t = 3 )$, but users can adjust it with minimal tuning cost.
>
>     For sparsity rate, Section 5.4 shows performance remains stable within a reasonable range, with degradation occurring only beyond a certain upper limit, which depends on molecular size. Users can select this parameter based on our experimental results for their specific molecular sizes.
>
> **Other Comments Or Suggestions:**
>
> 1. Thank you for pointing that out—the more precise shape should be $ n \times n_0 $, where $ n_0 $ corresponds to the number of occupied orbitals. The reason we sometimes write $ C $ as $ n \times n $ is that the KS equation is typically solved for all eigenstates, including virtual orbitals. We will revised this and running title in the revised version.
>
> **Questions For Authors**
>
> 1. Please see previous responses.
>
> 2. Thank you for raising this question. We addressed this in Section 4.1. To ensure that the combination selection is as unbiased as possible, we initialized the learnable matrix $ W $ (the sparsity weights) as an all-one vector. This means that, at the beginning, all combinations are considered to have the same importance.
>
> 3. Thank you very much for pointing this out. Here, the result of inner product is still a irrep with different order of feature, and the superscript stands for the order of irrep feature we used (actually, all the superscripts in our paper stand for the order of irreps feature). But we just found that there is a typo in the writing of the formula. The correct formula should be $<x_i,x_j>^{1:}$, and the superscript ${1:}$ stands for the irrep feature larger than order zero. We are very sorry about the mistake and have fixed it in the manuscript.

---

### Official Review · Reviewer_PzTP · 2025-03-28

**Overall Recommendation:** 4

**Summary:**

This paper introduces SPHNet, an SE(3) equivariant graph neural network designed to efficiently and scalably predict Density Functional Theory (DFT) Hamiltonian matrices. The core contribution is the incorporation of adaptive sparsity to address the significant computational cost associated with high-order tensor products (TP) operations in equivariant networks, which limits their application to large molecular systems. SPHNet employs two novel mechanisms: a Sparse Pair Gate to filter unimportant node pairs and a Sparse TP Gate to prune less significant interaction components within the tensor products. To manage the sparsity dynamically during training, a Three-phase Sparsity Scheduler (random, adaptive, fixed phases) is proposed to ensure stable convergence while achieving high sparsity levels (up to 70%). The main findings reported are that SPHNet achieves state-of-the-art accuracy on the QH9 and PubchemQH datasets while demonstrating significant computational improvements both in speedup and memory usage.

## Update after rebuttal

Thank you for your thoughtful rebuttal. I appreciate the clarifications provided regarding your methodology. I look forward to the revised manuscript.

**Claims And Evidence:**

The main claims are:
1.  SPHNet significantly improves computational efficiency (speed and memory) for Hamiltonian prediction compared to existing SE(3) equivariant models.
2.  This efficiency gain is achieved through novel adaptive sparsity mechanisms (Sparse Pair Gate, Sparse TP Gate, Three-phase Sparsity Scheduler) that reduce tensor product operations.
3.  SPHNet maintains or improves prediction accuracy despite the induced sparsity, achieving SOTA results on QH9 and PubchemQH datasets. The ablation study supports the claim that significant sparsity (up to 70% for PubChemQH) can be introduced without substantial accuracy loss.

The evidence provided appears generally supportive:
* Experimental results on benchmark datasets (QH9, PubchemQH, MD17) are presented, comparing SPHNet against baselines like QHNet and WANet.
* Quantitative results are reported, claiming up to 7x speedup and 75% memory reduction.

**Essential References Not Discussed:**

From the perspective of PIML for DFT, the paper covers the key related areas reasonably well, citing major works in SE(3) equivariance, Hamiltonian prediction, and general network sparsification.

**Experimental Designs Or Analyses:**

As stated in methods and evaluation criteria, the experimental design and analyses are reasonable, with good baselines, datasets, metrics and ablation studies.

**Methods And Evaluation Criteria:**

Yes, the proposed methods and evaluation criteria are appropriate for the problem of accelerating DFT Hamiltonian prediction using physics-informed machine learning.

* The core method involves introducing adaptive sparsity into an SE(3) equivariant GNN architecture. Targeting the tensor product operations, known computational bottlenecks in such networks, with sparsity is a reasonable strategy for improving efficiency. The specific gate mechanisms (Sparse Pair, Sparse TP) and the sparsity scheduler are novel contributions designed to implement this strategy effectively.
* Evaluation Criteria: The evaluation uses standard metrics:
    * MAE of the predicted Hamiltonian elements, observables and similarity of coeff matrix elements, compared to DFT calculations.
    * Inference speed (Samples/Sec) and GPU memory usage for assessing scalability, which is a primary goal of the paper.
    * Established datasets like QH9, PubchemQH, and MD17 are used. The larger basis set size of Def2-TZVP is important for showing scalability.

**Other Comments Or Suggestions:**

* SPHNet is a somewhat famous model used for 3D point cloud analysis (https://arxiv.org/abs/1906.11555). I think the domains are far enough that it can't be confused, but consider changing the name of the model.
* What do "higher coefficient rates" mean in this context? It is used to claim even more performance gains )Page 7, above Table 3).
* Discuss the computational overhead of the sparsity gates and scheduler themselves, although this is likely small compared to the savings from reduced tensor products.
* Minor:
    * Use the draft/review latex environment when submitting for review so that the line numbers are visible.
    * Page 6: Should be "The GTO orbital basis is used for MD17 and QH9  is used for ..."
    * Page 6: Note instead of Noted
    * Maintain consistent spacing for citations

**Other Strengths And Weaknesses:**

* The authors show that by using the TSS induced sparsity, the inference speed and memory usage of SE(3) network based Hamiltonian prediction is greatly improved while maintaining similar accuracy as baseline models.
*  The reported efficiency improvements (speed and memory) are substantial (up to 7x speedup, 75% memory reduction) while maintaining competitive or SOTA accuracy, demonstrating practical value. This is especially important for the PubchemQH results.
* The paper is generally well-written and clearly structured. The methodology is explained with helpful diagrams. The experiments are comprehensive and well-documented in the main text and appendices.


I am a bit confused about the notation used in equation (1). Shouldnt H and S be basis size dependent ($n \times n$), $\epsilon$ be orbital set size dependent ($n_o \times n_o$) and C be ($n \time n_o$)? I apologize if I misunderstood the notation.

**Questions For Authors:**

1. the selection probability for pairs *increases* with distance in the Sparse Pair Gate is counter-intuitive compared to typical distance cutoffs and goes against the nearsightedness principle (E. Prodan,& W. Kohn,  Nearsightedness of electronic matter, Proc. Natl. Acad. Sci. U.S.A. 102 (33) 11635-11638, https://doi.org/10.1073/pnas.0505436102 (2005)). Can you explain this further, and does this hold across different systems/basis sets?
2. What is the computational overhead introduced by the Sparse Pair Gate, Sparse TP Gate, and the adaptive phase of the scheduler relative to a dense model operating at the same FLOP count?

**Relation To Broader Scientific Literature:**

This paper fits within the active research area of applying geometric deep learning to directly predict the converged Hamiltonian under DFT. It improves the efficiency of SE(3) equavariant GNNs by reducing the tensor product operations needed by improving network sparsity. The effectiveness of this technique is shown on DFT Hamiltonian prediction given a molecule graph.

**Theoretical Claims:**

The paper focuses primarily on algorithmic innovation and empirical validation rather than presenting novel theoretical claims or proofs within ML or DFT. The effectiveness of the Three-phase Sparsity Scheduler seems justified empirically.

---

> ### Author Rebuttal · Authors · 2025-03-31
>
> Thank you very much for your valuable comments and suggestions. Below are our detailed responses.
>
> **Other Strengths And Weaknesses**
>
> Thank you for pointing that out—your notation is the more precise way to express it. $C$ should indeed be $n \times n_0$ in Equation 1, where $n_0$ corresponds to the number of occupied orbitals. We sometimes write $C$ as $n \times n$ because the KS equation is typically solved for all eigenstates, including virtual orbitals. We appreciate your keen attention to notation and will make sure to clarify this in the revised version.
>
> **Other Comments Or Suggestions**
>
> 1. Thank you for pointing this out. We are carefully considering this, and while we may not be able to provide a definitive new name in the short term, we will update this in the subsequent versions.
>
> 2. The higher coefficient rates imply that as the molecular system grows larger, we can apply a higher sparsity rate to further accelerate the model without sacrificing accuracy. We apologize for the confusion and have revised this to 'higher sparsity rates' for better clarity.
>
> 3. As you mentioned, compared to the tensor product operation, all these operations only introduce minimal computational overhead and have little impact on the overall speed. However, this discussion is still meaningful, and we will include it in the subsequent version of the manuscript.
>
>     In the three-phase sparsity scheduler, for a given unsparsified set $U$, the additional computational overhead in the first phase has a complexity of $\mathcal{O}(|U|)$, contributed by the RANDOM(·) operation. The second phase has a computational overhead of $\mathcal{O}(|U| \log{|U|})$, arising from the TOP(·) operation. Since we fix the learnable weight matrix and the selected elements, there is no additional computational overhead in the third phase. For detailed information, please refer to Equation 3.
>
>     For the sparse TP gate, the computational overhead comes from the element-wise multiplication of two weight vectors (Equation 5), so its complexity is $\mathcal{O}(|U_c|)=\mathcal{O}(L^3)$.
>
>     For the sparse pair gate, the additional computational overhead mainly comes from the linear layer $F_p(\cdot)$ in Equation 7, with its complexity being $\mathcal{O}(d_{hidden}|U_p|)$, where $|U_p|$ is always the square of the number of atoms and $d_{hidden}$ is the hidden feature dimension. Other operations, including the inner product (Equation 6) and the weight calculation (Equation 9), are necessary operations in our framework, even without the sparse pairwise gate.
>
> 4. -7. Thank you for the comments. We have revised the manuscripts accordingly.
>
> **Questions For Authors**
>
> 1. Thank you for the question. We agree with your point. In fact, in our experiments, the number of selected short-distance pairs still constitutes the majority of all selected pairs, which aligns with the nearsightedness of electronic matter, as shown in [Figure A](https://imgur.com/a/0EGNbF8). Each bar represents the fraction of selected pairs within the range from k to k+1 Å relative to the total number of selected pairs. Therefore, when we say *“as the pair length increases, the probability of a pair being selected also rises,”* we mean that pairs at longer distances are indeed retained at a higher proportion in [Figure B](https://imgur.com/a/0EGNbF8)—though their absolute count remains significantly lower compared to short-distance pairs. For example, among 10 pairs at 25–26 Å, 6 may be selected, whereas among 300 pairs at 4 Å, 100 may be selected.
>
>     There are two possible reasons for the observed tendency to retain long-distance pairs. First, long-range interactions, including electrostatic interactions and weak van der Waals forces, are crucial for accurately describing large molecules. Consequently, incorporating such interactions could be beneficial for overall accuracy, a property that has been studied in previous research (Knörzer J,  et al. https://arxiv.org/pdf/2202.06756; Li Y, et al. https://arxiv.org/pdf/2304.13542.). Second, since long-distance pairs constitute only a small fraction of all pairs, once sufficient data has been collected to characterize short-range interactions, selecting more long-distance "hard samples" could contribute to improving final accuracy.
>
>     However, due to the lack of large molecular system datasets, we conducted a similar experiment on the QH9 dataset (Figure A-B), where we found that the selected ratio was similar to the proportion of short-distance pairs in the PubChemQH dataset. However, since the maximum atomic distance in this dataset is less than 8 Å, we did not observe a preference for retaining long-distance pairs. This is reasonable, as long-range atomic interactions typically become significant only when the interatomic distance exceeds 12 Å. As larger molecular datasets become available in the future, we hope to conduct further experiments to validate this hypothesis.
>
> 2. We have discussed that in the previous responses.

---

> > ### Comment · Reviewer_PzTP · 2025-04-07
> >
> > Thank you for your thoughtful rebuttal. I appreciate the clarifications provided regarding your methodology. I look forward to the revised manuscript.

---

### Decision · Program_Chairs · 2025-05-01

**Decision:**

Accept (poster)

**Comment:**

This paper proposes SPHNet, an efficient SE(3) equivariant graph neural network for Hamiltonian matrix prediction. By introducing sparse gating mechanisms and an adaptive sparsity scheduler, SPHNet effectively reduces computational costs while maintaining high accuracy. Experiments on two common datasets demonstrate that SPHNet achieves state-of-the-art performance with significant speed and memory improvements.

The reviewers were all recommending acceptance of the paper.